# Adaptation Strategies of Halophytic Barley *Hordeum marinum* ssp. *marinum* to High Salinity and Osmotic Stress

**DOI:** 10.3390/ijms21239019

**Published:** 2020-11-27

**Authors:** Stanislav Isayenkov, Alexander Hilo, Paride Rizzo, Yudelsy Antonia Tandron Moya, Hardy Rolletschek, Ljudmilla Borisjuk, Volodymyr Radchuk

**Affiliations:** 1Leibniz-Institute of Plant Genetics and Crop Plant Research (IPK), Corrensstrasse 3, 06466 Gatersleben, Germany; hilo@ipk-gatersleben.de (A.H.); rizzo@ipk-gatersleben.de (P.R.); moya@ipk-gatersleben.de (Y.A.T.M.); rollet@ipk-gatersleben.de (H.R.); borisjuk@ipk-gatersleben.de (L.B.); 2Institute of Food Biotechnology and Genomics NAS of Ukraine, Osipovskogo Street, 2a, 04123 Kyiv, Ukraine

**Keywords:** halophytic wild barley, salinity, osmotic stress, metabolome, transcriptome, ionome, stress adaptation, *Hordeum marinum*

## Abstract

The adaptation strategies of halophytic seaside barley *Hordeum marinum* to high salinity and osmotic stress were investigated by nuclear magnetic resonance imaging, as well as ionomic, metabolomic, and transcriptomic approaches. When compared with cultivated barley, seaside barley exhibited a better plant growth rate, higher relative plant water content, lower osmotic pressure, and sustained photosynthetic activity under high salinity, but not under osmotic stress. As seaside barley is capable of controlling Na^+^ and Cl^−^ concentrations in leaves at high salinity, the roots appear to play the central role in salinity adaptation, ensured by the development of thinner and likely lignified roots, as well as fine-tuning of membrane transport for effective management of restriction of ion entry and sequestration, accumulation of osmolytes, and minimization of energy costs. By contrast, more resources and energy are required to overcome the consequences of osmotic stress, particularly the severity of reactive oxygen species production and nutritional disbalance which affect plant growth. Our results have identified specific mechanisms for adaptation to salinity in seaside barley which differ from those activated in response to osmotic stress. Increased knowledge around salt tolerance in halophytic wild relatives will provide a basis for improved breeding of salt-tolerant crops.

## 1. Introduction

High salinity is one of the biggest threats to modern agriculture and crop productivity, leading to an annual estimated economic loss of over 10 billion USD [1]. More than 800 million hectares of agricultural land (>6% of the planet’s total land area) are considered to be salt-affected [2]. The area of salinized soils is reported to be increasing at a rate of 10% per year, and is an issue in more than 100 countries worldwide [3,4].

High levels of salinity result in impaired plant growth and development through various mechanisms, including osmotic stress (OST) due to loss of cellular water content, cytotoxicity due to excessive uptake of Na^+^ and Cl^−^ ions, oxidative stress due to generation of reactive oxygen species (ROS), and nutritional imbalance [5]. Compared to salt-sensitive plants, or glycophytes, the increased salt tolerance of plants grown in a saline environment, or halophytes, is achieved predominantly by a greater robustness of employed mechanisms rather than qualitative differences [5,6]. These mechanisms involve maintaining the homeostasis of cellular ions, making osmotic adjustments and ROS scavenging. Na^+^ and Cl^−^ ions are themselves the important contributors to the cellular osmotic potential [7]. Because they are toxic if not compartmentalized, these ions have to be sequestrated into vacuoles or endosomal compartments by ion exchangers and the H^+^ pumps localized to the tonoplast or endosomal membranes [8]. Organic osmolytic solutes, such as sugars, sugar alcohols, and proline, accumulate in the cytoplasm of halophytic species to balance the osmotic potential of Na^+^ and Cl^−^, contained in the vacuole, and to maintain the physiological functions of the cell [9]. From an energy-saving aspect, cellular osmotic adjustment is achieved more efficiently by the use of ions than of organic solutes [7].

Plant species have evolved diverse and unique ways to survive in harsh saline environments. Certain dicot halophytic plants, in order to resist or avoid accumulation of toxic ions, have developed special structures and organs, such as epidermal bladder cells, which accumulate excessive Na^+^ in their vacuoles, and hydathodes, which actively secrete salt and reduce the concentration of toxic ions in the cells [10,11]. The majority of halophytic monocots do not exhibit such specialized organs, but have developed other ways to survive under saline conditions. Several wild species within the Triticeae tribe, to which the major crops wheat (*Triticum aestivum*) and barley (*Hordeum vulgare*) belong, exhibit exceptional salinity tolerance [10]. The seaside barley (*Hordeum marinum*), a typical Mediterranean halophytic plant of coastal salt marshes, is considered one of the major genetic sources for salinity tolerance [12]. The amphidiploid wheat hybrids with *H. marinum* exhibit improved salt tolerance compared with wheat [13,14]. *H. marinum* possesses a higher water saturation deficit and osmotic potential in comparison with that of cultivated barley due to higher accumulation of proline, glycine betaine, and dehydrins [15,16]. Proteomic analysis also revealed increased levels of proteins involved in energy metabolism [15]. Furthermore, antioxidant enzymes in seaside barley were shown to have significantly higher activity in plants grown at high salinity [17]. Transcriptome studies suggest that the salt-tolerance strategy of *H. marinum* comprises low energy consumption, utilization of inorganic ions as cheap osmotic agents, and changes in the activity of the HmHKT1;5 and HmHKT2;1 transporters [18,19,20]. However, the molecular mechanisms underlying the biochemical and morphological changes and physiological strategies employed by *H. marinum* during acclimation to salinity remain mostly unexplored.

The aim of the present study was to elucidate the differences in adaptation strategies of *H. marinum* plants to OST and salinity stress (SST) at the molecular, metabolic, morphological, and physiological levels.

## 2. Results

### 2.1. Different Physiological Responses of H. marinum and H. vulgare to OST and SST

To evaluate differences in the response of *H. vulgare* and *H. marinum* to SST and to elicit salinity adaptation responses in halophytic seaside barley, the plants were cultivated in hydroponic culture containing 300 mM NaCl, which corresponds to slightly over 500 mOsm osmotic pressure. Preliminary OST experiments showed deleterious effects on plants after treatment with the same osmotic pressure (32% PEG6000), probably due to impermeability of this osmotic agent through cell membranes. Therefore, plants were cultivated in media supplemented by 15% PEG6000, which plants could still tolerate. SST and OST treatments affected the growth of both *H. vulgare* and *H. marinum* plants, albeit to a different extent (Figure 1). The relative growth rate (RGR) of *H. marinum* plants was approximately 2-fold lower under either SST or OST conditions (Figure 1C). A similar decrease in the RGR of *H. vulgare* plants was observed under OST; however, application of SST resulted in a ~95% reduction of growth rate relative to the control (Figure 1F). Under high salinity, the *H. vulgare* plants exhibited leaf chlorosis and wilting as marks of severe salt toxicity, whereas the *H. marinum* plants maintained their strong green color (Figure 1A,B). Under OST, however, they turned a yellow shade. Furthermore, after SST, the osmotic pressure recorded in *H. marinum* plants was slightly lower than that in *H. vulgare* (Appendix A).

Comparative non-invasive magnetic resonance imaging of *H. marinum* plants demonstrated alterations in the hypocotyl and root structure, compared to the controls, under SST (Figure 2). Numerous root primordia and seminal roots were initiated in the hypocotyl region, resulting in more fibrous roots (Figure 2A,B). NMR models (Figure 2C,D) allowed the calculation of volumes and surface area of roots. While the volumes of individual roots were only marginally decreased under conditions of high salinity compared to the control (0.43 ± 0.20 vs. 0.55 ± 0.10, mm^3^), the total surface area of the stressed roots was ~23.8% higher, due to the production of a larger number of thinner roots.

Plant water content (PWC) in *H. marinum* tissues was depleted by ~30% after application of either OST or SST (Figure 1D). In particular, the cortex region of the saline-affected roots contained less water than that of the control roots (Figure 2E,F). A decrease of ~25% in PWC was also observed in the *H. vulgare* plants under OST, and SST resulted in almost 50% less PWC compared to the control plants (Figure 1G). Thus, seaside barley exhibited a greater ability to retain water under conditions of high salinity in comparison to *H. vulgare*. Finally, a greater reduction in shoot:root weight ratio was observed in *H. vulgare* plants under SST than in *H. marinum* (Figure 1E,H). Together, these data indicate that stress treatments, in particular salinity, hinder the growth of *H. vulgare*, while *H. marinum* exhibits stronger resistance to SST, as reflected in enhanced water retention, preserved shoot growth, and, possibly, sustained biosynthetic activity.

### 2.2. Different Photosynthetic Activity and Assimilate Allocation in H. marinum and H. vulgare Plants Under SST

To evaluate the photosynthetic activity and assimilate allocation in *H. marinum* and *H. vulgare* plants under stress conditions, we analyzed the uptake and distribution of assimilates following the treatment of control and stressed shoots with ^13^C-labeled CO_2_ (Figure 3). When compared with domesticated barley, *H. marinum* shoots showed ~2-fold higher efficiency of ^13^C uptake. In *H. marinum*, the efficiency of ^13^C assimilation was slightly decreased under OST, and not significantly changed under SST, indicating maintenance of photosynthetic activity rate. On the other hand, the ^13^C assimilation in *H. vulgare* shoots appeared significantly decreased under OST and was almost negligible under SST (Figure 3A).

The *H. marinum* control plants re-allocated large amounts of ^13^C-labeled assimilates to the roots (Figure 3B). Application of either OST or SST led to a significant decrease, but not a complete block of the ^13^C allocation to the roots. The *H. vulgare* roots also accumulated less ^13^C-labeled assimilates than the equivalent control plants under OST, while ^13^C accumulation was barely detectable in SST-treated roots due to the inhibited photosynthetic ^13^C fixation by the shoots.

### 2.3. Comparative Analysis of Mineral Composition Under SST and OST

We analyzed the mineral content of *H. marinum* roots and shoots under control and stress conditions (Table 1).

Following incubation with 300 mM NaCl, a marked elevation of Na content was observed in both tissue types, albeit ~1.4-fold lower in the shoots than in the roots. Contrary to Na, the K and particularly the Ca contents were significantly reduced in the shoots and roots of SST-treated plants. Ca^2+^ is recognized as a crucial second messenger in signaling pathways linking the perception of environmental stimuli to plant adaptive responses [21]. The estimated K/Na ratio was higher in the shoots than in roots (1.66 vs. 0.75) under SST, possibly indicating more efficient K^+^ retention in green tissue. OST led to K reduction but Ca elevation in the roots, whereas no change was observed in their levels in the shoots. Zn and Mo contents were also elevated in both tissues under SST and in the shoots under OST. In OST-treated roots, Zn content was reduced, while Mo was not affected. Furthermore, under SST, P and Mg concentrations were decreased in both sample types, and S and B contents were decreased only in the shoots. These results suggest that salinity evokes changes in mineral uptake and allocation in the whole plant to counteract Na and Cl excess and adjust the osmotic pressure. In contrast to SST, OST resulted in a rise in mineral contents (P, Mn, Ni, and S) in the shoots, accompanied by decreases in the Cu and Na contents in the roots, likely reflecting ionic adaptations to high osmotic pressure.

### 2.4. Alterations of Metabolite Profiles in Response to SST and OST

Changes in metabolome of the roots and shoots of *H. marinum* plants were investigated under OST and SST by untargeted metabolite profiling. In total, 138 and 136 metabolites were identified in the roots and shoots, respectively (Appendix A). In the roots, the levels of 59 metabolites were significantly altered by >2-fold (a decrease observed in 44 and an increase in 15 metabolites) following treatment with OST, whereas a change was detected in 61 metabolites (35 decreased and 26 increased) in those under SST conditions. In the shoots of the plants, OST led to a change in the levels of 68 metabolites (45 decreased and 23 increased) and SST resulted in differences for 58 metabolites (40 decreased and 18 increased) (Appendix A).

Principal component analysis of the metabolite profiles of *H. marinum* plants revealed differential responses to OST versus SST (Appendix A), with only 14 metabolites being affected under both stresses. The largest increase was detected for the flavonoid 3-methoxy-4-hydroxyhippuric acid under both types of stress (Appendix A). The levels of ascorbate and its precursor mannose-6P were increased under OST but decreased under SST in the shoots (Figure 4). A strong increase in gluconolactone, a polyhydroxy acid with metal-chelating and ROS-scavenging activities [22], was specifically detected in SST-treated roots (Appendix A). These data suggest differences in ROS production and scavenging in *H. marinum* tissues under OST versus SST.

Plants often exhibit an increase in free proline following exposure to hyperosmotic stress or SST [23]. In *H. marinum*, such an increase was larger under SST than OST in both tissue types. The trehalose-6P content was decreased in both tissues under OST, while an increase in mannitol-1P was observed in the roots, but not the shoots, under both stresses (Figure 4). These elevated levels indicate increased mannitol biosynthesis.

The levels of fructose-6P and lactate were decreased in both tissues under OST and SST conditions (Figure 5). An increase in citrate in both tissues under OST and in shoots under SST, but decrease in organic acids associated with malate conversion [2-oxoglutarate, succinate (in both tissues after SST and in roots after OST), fumarate (only in shoots) and malate], were detected in the metabolites associated with the tricarboxylic acid (TCA) cycle (Figure 5). The levels of purine nucleotides generally exhibited trends opposite to those of uric acid, allantoin, and allantoate in roots and shoots, particularly under OST (Figure 6). These results indicate that, in stressed plants, there is a reduction of processes related to nucleotide and energy metabolism and possibly cellular proliferation.

Homeostasis of the auxin indole-3-acetic acid (IAA) is achieved through amino acid conjugation and catabolism [24]. The levels of the IAA-Ala conjugate (reversible storage compound of IAA) were increased in plants under both type of stress. 2-oxindole-3-acetic acid (oxIAA), a major inactive and irreversible IAA degradation product, was markedly increased in SST-treated roots (Appendix A). Thus, adjustment of auxin content may be involved in salinity adaptation.

Levels of mevalonate pyrophosphate, involved in the mevalonate pathway of terpenoid backbone biosynthesis [25], were decreased in all tissues, particularly in roots under SST (Appendix A). The content of methylerythritol-4P (MEP), part of the MEP pathway of terpenoid synthesis, decreased in shoots under both types of stress (Appendix A). Distinct components of the shikimate pathway (e.g., quinic acid and shikimate-3P) were significantly reduced under OST. Tyramine content, a product of tyrosine metabolism and a precursor in alkaloid biosynthesis via the shikimate pathway, was decreased in the roots under both types of stress (Appendix A).

### 2.5. Transcript Profiling in H. marinum Suggests a Stronger Influence by OST than SST

We analyzed changes in RNA transcript abundances in the roots and shoots of *H. marinum* following treatment with OST and SST, using RNA sequencing. In total, 2232 differentially expressed genes (DEGs) with known or predicted function were detected in at least one tissue type as a result of at least one type of stress (fold change in expression ≥ 3, false discovery rate (FDR) < 0.05) (Appendix A). The largest changes in transcriptome were observed in the OST-treated shoots with 1210 DEGs detected (821 down- and 389 upregulated), followed by the OST-treated roots with 1063 DEGs (469 down- and 594 upregulated). Notably fewer DEGs were detected under SST in the roots (545 DEGs: 220 down- and 325 upregulated) and shoots (270 DEGs: 106 down- and 164 upregulated genes). While a difference in DEGs between stressed roots and shoots was expected due to their functional specificity, the overlap between DEGs detected following OST versus SST of the same tissue was also small (Figure 7A,B).

### 2.6. OST Differentially Affects Shoot and Root Development

Of the DEGs detected in OST-treated shoots, the majority (117 genes) encoded proteins involved in transcription, translation, and general cellular processes, and included diverse *histones* (43 genes), *ribosomal proteins* (25 genes), *cyclins* (6 genes), *cell division control* (2 genes), and *expansins* (2 genes) (Appendix A); all of them were downregulated. The same group of DEGs was also prominent in OST roots (59 genes), in which the genes involved in control of cell division and elongation (*cortical cell-delineating proteins*, *expansins*, *mitogen-activated protein kinase*, *cell cycle control phosphatase*) were also repressed. Under OST conditions, the most highly upregulated gene in both the shoots and roots encoded rRNA N-glycosidase, which is involved in ribosomal degradation [26]. Three *pumilio* genes, whose products may influence mRNA stability [27], were also highly upregulated in the shoots. These data indicate inhibition of cell division and elongation in plants under OST. Furthermore, repression of *actin depolymerizing factor* and *actin* together with activation of *dynein*, two *β-tubulins*, and *flotillin-like protein* genes suggests reorganization of the cytoskeleton and intracellular trafficking under these conditions.

OST resulted in downregulation of genes associated with cell wall metabolism (cellulose synthases, fasciclin-like arabinogalactan proteins, pectinesterases, xyloglucan endotransglucosylases and UDP-glycosyltransferases) in shoots and roots, signifying strong repression of cell wall biosynthesis. In the roots, genes involved in lipid biosynthesis were mostly repressed (including 3-ketoacyl-CoA synthases and fatty acid desaturase), while those responsible for lipid degradation (GDSL esterase/lipases, papatins, and lipoxygenase) exhibited increased transcription. In the shoots, from the repression of GDSL esterase/lipases (9 genes), 3-ketoacyl-CoA synthases (6 genes), bifunctional inhibitor/lipid-transfer protein/seed storage 2S albumin superfamily proteins (6 genes), glycerol-3-phosphate acyltransferases (4 genes), fatty acid desaturases (3 genes), fatty acid hydroxylases (2 genes), lipoxygenases (2 genes), and phospholipases (2 genes), it can be deduced that total lipid metabolism was likely minimized.

In contrast to the response under OST, SST-treated plants revealed only minor transcriptional changes in genes involved in general cellular processes and in the metabolism of lipids and the cell wall (Appendix A). Downregulation of *expansins*, GTPase *RsgA*, CRIB domain-containing protein *RIC1,* and *α-tubulin 4*, as well as upregulation of *flotillin* and *β-tubulin 2*, indicate cytoskeleton reorganization in roots under SST. Of 20 DEGs associated with cell wall metabolism, three *pectin lyase* genes, two *fasciclin-like arabinogalactan protein* genes, and one *xyloglucan endotransglucosylase* gene showed decreased expression, while four *UDP-glycosyltransferase* genes and one *xylanase inhibitor* gene revealed increased expression in the roots. Moreover, the transcription of five genes encoding laccase, which is involved in lignin biosynthesis, was increased up to 10-fold in both the shoots and roots, suggesting increased lignification of cell walls under SST conditions.

### 2.7. OST But Not SST Leads to Strongly Diminished Photosynthetic Processes

OST led to a marked downregulation of genes encoding proteins from the entire photosynthetic machinery (Table 2): *chlorophyll a-b binding proteins* (18 genes), *subunits of reaction center of photosystems I and II* (9), *thylacoid membrane proteins* (7), *RUBISCO* (4), *subunits of cytochrome b6-f complex* (2), *plastocyanin*, *ferrodoxin,* and *ribulose-5P-3-epimerase*. However, this was not observed under SST conditions. *Heme oxygenase*, whose product plays a role in the protection against oxidative damage via ROS scavenging [28], was upregulated in shoots under both OST and SST. The expression of *protein D1*, required for the repair of photosystem II [29], was increased over 44-fold in SST-treated shoots.

### 2.8. Primary Metabolism and Sugar Conversion Are Altered Under SST and OST

Upregulated expression of genes involved in sucrose cleavage (*sucrose synthases* and *invertases*) as well as starch and glucan degradation (*glucan-1,3-β-glucosidases,* and *β-amylase*), together with the reduction of expression of those responsible for fructan biosynthesis (*sucrose:sucrose 1-fructosyltransferase*, *sucrose:fructan 6-fructosyltransferase,* and *fructan:fructan 1-fructosyltransferase*), suggest a shift from the production of di- and polysaccharides towards their degradation to hexoses in plants under both types of stress (Figure 5). The transcription of *aldose reductase* and *α-galactosidase*, involved in monosaccharide conversion and sorbitol synthesis, was also increased, as was the expression of *trehalose-P synthase* and *trehalose-6P phosphatase* was also upregulated in the roots (Appendix A).

A group of DEGs associated with the TCA cycle and glycolysis showed upregulation in OST-treated plants (Table 2). However, the expression of *pyruvate dehydrogenase* and *NAD-dependent malic enzyme* was also increased in plants under SST (Figure 5). Increased transcription of *isocitrate lyase* and *malate synthase* suggests activation of the glyoxylate bypass [30,31]. The expression of *alanine:glyoxylate aminotransferase* and *NADP-dependent malate dehydrogenase*, whose products are involved in photorespiration, was increased only in plants treated with OST (Appendix A).

*Allantoinase*, whose product converts allantoin into allantoate, was repressed in shoots under both stress conditions, while *AMP-deaminase* expression was upregulated, suggesting activation of metabolic conversion of adenine ribonucleotides into allantoin. Similarly, *adenine phosphoribosyltransferase 5*, responsible for *de novo* synthesis of adenosine monophosphate (AMP), was upregulated under OST, whereas *adenylate kinase*, which performs interconversion of adenine nucleotides, was downregulated highlighting a channeling of adenine nucleotides towards catabolism (Figure 6).

### 2.9. SST and OST Affect the Expression of Distinct Groups of Transporters

*H. marinum* roots under both OST and SST demonstrated marked changes in expression of genes encoding different membrane transport proteins, including ion and anion transporters (Table 2), the majority of which were upregulated (Appendix A). The pattern of salinity-responsive DEGs related to membrane transport, was mostly different to that resulting from OST.

Among anion transporters, three *boron transporter* genes with potential anion efflux activity [32] were specifically and highly upregulated in SST-treated roots, suggesting a role for them in Cl^−^ removal. Upregulation of S-type anion channels *SLAH2* and *SLAH3* may serve the purpose of enrichment with NO_3_^–^, as a main competitor of Cl^−^, to minimize Cl^−^ accumulation [33,34]. In the shoots of SST-treated plants, upregulation of *NRT1/PTR FAMILY 7.3*, a potential anion transporter [35], may be linked to further regulation of root-to-shoot anion transport. In OST-treated roots, however, six other genes from the *NRT1/PTR* family were downregulated. Increased expression of an aluminum-activated *malate transporter* may also be associated with Cl^−^ efflux [36,37] or malate extrusion into soil to increase phosphate availability [38]. In line with the latter possibility, five *phosphate transporters* were strongly upregulated in the roots under SST.

Among ion transporters, the expression of K^+^ channel *SKOR* was increased ~11-fold exclusively in SST-treated roots, indicating enhanced re-translocation of K^+^ as the main Na^+^ competitor [39,40]. The transcript levels of *cation/H^+^ antiporter 16,* another potential player in the maintenance of Na^+^/K^+^ homeostasis, were also increased. The expression *HKT5* transporters and *NHX* exchangers, shown to be important for adaptation to high salinity [41], was unchanged in *H. marinum* under SST, but decreased under OST (*HKT14;1* and *HKT1;1* in the shoots, *HKT1;5* and *HKT2;1* in the roots). Upregulation of *ammonium transporter 2* in roots under both stress types may indicate increased ammonium transport to foster N demand in stressed plants. Ammonium assimilation is less energy-demanding than nitrate uptake [42]. In line with this, six genes encoding high-affinity nitrate transporters were repressed in roots under OST, further supporting the hypothesis of increased ammonium uptake. OST, but not SST, led to the marked amplification of the expression of seven Zn transporter genes (up to 8-fold), five plant *cadmium resistance protein* (PCRP) genes, two *Zn-facilitator like protein* (ZFLP) genes, and two *YELLOW STRIPE-like proteins* (*YSL*) genes. ZFLP1 participates in polar auxin transport and drought stress tolerance in *Arabidopsis* [43]; PCRPs are involved in both Zn extrusion and long-distance transport [44]; and YSLs are thought to be implicated in the transport of metals [45]. Two glutamate receptors, *GLR1.3* and *GLR2.8*, both nonselective cation channels [46], were upregulated in OST-treated roots and shoots, respectively. *Mechanosensitive ion channel 10* (*MSL10*) was downregulated in roots under both types of stress. Membrane tension during stress may lead to activation of cation conductance via MSL channels [46]. Moreover, two cyclic nucleotide-gated channels, which may also be involved in Na^+^ uptake [47], were downregulated ~10-fold in roots under both stresses.

Different sucrose exporter genes SWEET [48] were upregulated in tissues under particular stress treatments: *SWEET12* and *SWEET14b* in roots under both stresses, *SWEET13a* and *SWEET13b* in SST-treated roots, and *SWEET14a* and *SWEET15b* in OST-treated plants. However, hexose transporter genes *SWEET2b* and *SWEET16* were downregulated in shoots under OST. Two monosaccharide transporters were upregulated under SST. Furthermore, *GDP-mannose transporter 1* was upregulated in both tissue types following both stresses. Of 14 DEGs encoding aquaporins, which mediate water uptake and movement in plants, 12 were repressed in at least one of the two tissues under at least one type of stress.

### 2.10. Gene Expression Analysis Suggests Differences in Amino Acid and Secondary Metabolite Accumulation Under OST and SST

Regarding DEGs related to amino acid metabolism, the majority were upregulated in SST-treated plants (Appendix A). High expression of genes involved in phenylalanine, tyrosine, and tryptophan metabolism (*tyrosine decarboxylase*, 10 genes; *anthranilate synthase*, 2 genes; *tryptophan synthase*, 2 genes; and *anthranilate phosphoribosyltransferase*) indicate a shift towards alkaloid, diterpene, and phenylpropanoid biosynthesis. Transcripts of *proline dehydrogenase 2*, involved in proline degradation, were decreased, but those of *prolyl 4-hydroxylase*, involved in proline synthesis, were increased, in line with proline enrichment in SST-treated plants (Figure 4). Repressed genes were mainly associated with amino acid degradation (*γ-glutamyl P-reductase*, 2 genes; *glutamate decarboxylase*; *choline dehydrogenase*, 2 genes; *cystathionine β-lyase*, and *phenylalanine ammonia lyase*). An increase in *glutamate synthase* transcripts indicates enhanced synthesis of glutamate, which can regulate ion transport via selective glutamate-gated cation channels [49].

In roots under both types of stress, increased expression of *isovaleryl-CoA-dehydrogenase* and *copalyl-diP synthase*, which use amino acid degradation products to produce diterpenoids, and two *sterol C4-methyl oxidase* genes, further support a shift towards diterpenoid biosynthesis. From the downregulation of two *cinnamoyl-CoA reductase* genes, two *phytoene synthase* genes, one *β-carotene hydroxylase,* and one *β-carotene isomerase*, it appears that the synthesis of isoprenoids and carotenoids may be hampered under OST. The expression of *ascorbate oxidase*, encoding an ascorbate-degrading enzyme, was increased up to 14-fold in roots under OST and SST. Strong upregulation of 12 *nicotianamine synthase* genes was observed specifically in OST-treated shoots and roots.

### 2.11. Gene Expression Analysis Indicates Changes in Plant Hormone Levels Under Stress

Both types of stress-induced changes in the expression of genes responsive to abscisic acid (ABA), the key phytohormone in adaptation to stress [50]. Strong transcriptional upregulation (up to 12-fold) of genes encoding ABA-induced small hydrophilic proteins [50], together with elevated expression of the ABA-responsive *GRAM domain-containing protein* genes [51], imply an increased ABA level in the roots under both stress conditions. However, downregulation of two *ABA receptor PYR1* genes only in SST-treated roots indicate differences in ABA perception and signaling under the two types of stress.

Strong transcriptional activation in SST-treated roots (up to 12-fold) of two *indole-3-glycerol phosphate synthase* genes, encoding a branch-point enzyme in the tryptophan-independent IAA biosynthetic pathway, points to increased auxin synthesis under conditions of high salinity. Different genes encoding auxin efflux carrier proteins, involved in auxin transport, were upregulated in roots and shoots under both stresses. By contrast, the genes encoding the auxin-responsive proteins IAA23 and IAA4 were downregulated. IAA4 forms part of a module that negatively regulates adventitious root development in *Populus* [52].

Transcripts of *1-aminocyclopropane-1-carboxylate oxidase* and *jasmonate O-methyltransferase,* which encode the key enzymes in ethylene and jasmonate biosynthesis, respectively, were increased in plants under OST, whereas those of diverse *ethylene-responsive transcription factor* genes were decreased in roots under both stresses.

In line with lower Ca^2+^ content in SST-treated plants (Table 1), the transcript levels of three *calmodulin* genes, one *Ca^2+^-dependent protein kinase,* and one *Ca^2+^-sensing receptor* were decreased in SST-treated roots.

### 2.12. Overlap in Expression of Stress-Responsive Genes

Of the 44 stress-related DEGs detected in OST-treated roots, 28 were also detected in roots treated with SST (Appendix A). Of these, ten *dehydrin* genes were upregulated under both stresses. Between one (under SST) and eight (under OST) chaperone *DnaJ* genes were also upregulated. While the main function of dehydrins and chaperones is to protect biomolecules, certain dehydrins possess metal-binding capacity and are regarded as ROS scavengers [53]. Regarding gene expression in the shoots, 71 stress-related DEGs were detected under OST, and only 19 under SST, but with an overlap of 13 common DEGs. At least one gene encoding a proline-rich protein was downregulated in a particular tissue, possibly reflecting a redirection of proline into the free pool as a key stress-protecting amino acid. One group of pathogen-related genes (*chitinase*, 9 genes; *germin-like protein*, 3 genes) exhibited increased expression under one or both types of stress, while another (*disease resistance protein*, 4 genes; *defensin*, 2 genes) revealed a decrease. Six *thaumatin* genes were upregulated in OST-treated roots, three of which were also activated under SST. Two genes encoding kiwellin, a protein accumulated to high levels under SST in *H. vulgare* [54], were also upregulated in *H. marinum* under similar conditions. *Rapid alkalinization factor 23*, involved in regulation of salt tolerance in *Arabidopsis* [55], was upregulated in SST-treated shoots and roots. Various repeat domain protein families are also anticipated to be involved in abiotic stress [56]. Expression of a *Kelch repeat-containing protein* was increased in both tissues under the two types of stress, while *WD40* (6 genes), *pentatricopeptide* (7 genes), *tetratricopeptide* (5 genes), and *ankyrin repeat proteins* (2 genes) were additionally upregulated in OST-treated shoots.

In plants, stress generally induces the production of toxic ROS. Alterations in the expression of the *peroxidase* gene superfamily, whose products are involved in ROS scavenging, and lignin production, were observed: nine genes were downregulated while another eight were upregulated in roots under SST and, in part, OST. In the shoots, a total of 25 *peroxidases* were repressed, while only four genes were upregulated under OST, and four genes were downregulated and three were upregulated under SST. Rearrangements were also revealed in the expression of the *thioredoxin* family in plants under OST, but not SST. The increase in expression of two *catalase* genes and *Neighbor of BRCA1 gene 1*, involved in pexophagy [31], was detected only in SST-treated roots. Two genes encoding nonsymbiotic phytoglobin, a NO sensor involved in hypoxia response [57], were strongly upregulated in roots under both stresses. Of genes encoding glutathione S-transferase, which produces a strong nonenzymatic antioxidant glutathione implicated in abiotic stress tolerance, four were upregulated in SST-treated roots and ten in OST.

## 3. Discussion

While the growth and development of both *H. marinum* and *H. vulgare* plants were affected by OST and SST, domesticated barley plants suffered much more strongly, particularly from SST, as manifested by their leaf chlorosis and wilting, as well as by an almost complete halt in photosynthetic activity and assimilate transport. *H. marinum* plants, however, remained dark-green under SST. They had not even revealed higher photosynthetic efficiency under control conditions but were also able to maintain photosynthetic activity and carbon fixation under high salinity. Genes involved in the defense of the photosystem were transcriptionally boosted in SST-treated *H. marinum* shoots. These observations indicate a more efficient photosynthetic apparatus in *H. marinum* which deserves more detailed investigations in the future. As a result, *H. marinum* plants exhibited better growth capacity, water retention, and shoot development under SST. By contrast, the plants turned yellow under OST, even though exposed to 2.5-fold lower osmotic pressure as compared to SST, likely due to a significant stress response, also supported by observed changes in metabolite and transcript profiles. Genes of the photosynthetic machinery and chlorophyll metabolism were strongly repressed in OST-treated shoots, in line with the yellow color. Transcriptional inhibition of processes related to cell proliferation and differentiation, and activation of lipid and cell wall degradation processes appeared to be triggered predominantly by OST rather than SST. This was in agreement with decreased purine and pyrimidine levels, as well as increased levels of their degradation products under OST. From the fact that a larger number of genes associated with ROS detoxification were upregulated, OST likely caused more severe ROS production than SST. While upregulation of genes involved in the TCA cycle and, in part, glycolysis, coupled with decreased levels of glycolytic intermediates indicated increased energy metabolism during both types of stress, the plant response to OST was possibly more energy demanding than to SST. OST-treated plants likely utilize the glyoxylate cycle for additional energy production. These data are in line with the increased levels of proteins involved in energy metabolism [15]. Overexpression of SWEET proteins is indicative of increased sucrose transport to the roots, and its utilization for energy retrieval. Increased sugar degradation to hexoses, representing the main energy source, is further feasible. A significant increase in glucose content in *H. marinum* plants under salinity has been described recently [18,58].

Seaside barley possesses a higher capacity than domesticated barley to regulate osmotic homeostasis under SST. Differently to OST, where impermeable PEG6000 caused strong stress response, halophytic *H. marinum* might recruit Na^+^ and Cl^+^ ions under high salinity to regulate osmotic pressure with fewer energy investments for the plant. In addition, the biosynthesis and accumulation of other osmolites may further contribute to osmoregulation. This is in line with the higher proline levels and increased expression of genes encoding hydrophilic dehydrins, germin-like, and other osmolytic proteins detected in SST-treated plants. Additionally, mannitol-1P content was strongly increased in roots, especially under SST. Similarly, halophytic *Prosopis strombulifera* accumulate large amounts of mannitol-1P as an osmoprotective agent [59]. From the upregulation of *trehalose-P synthase* and *trehalose-6P phosphatase*, and decreased levels of trehalose-6P intermediate, SST-treated roots appear to accumulate trehalose, another known osmoprotective agent [60]. Moreover, as follows from the activation of genes of sorbitol synthesis, sorbitol accumulation is also possible. In tomato plants, increased aldose reductase activity and sorbitol synthesis were shown to improve salt tolerance [60]. Enhanced ureide accumulation may be associated with osmoprotection, but also to stabilization of proteins and membranes [61], efficient N utilization due a low C/N ratio of heterocyclic molecules that optimize the transport of organic N under reduced photosynthetic capacity [62], and activation of ABA and jasmonic acid signaling [63].

NMR analysis revealed morphological changes in the *H. marinum* roots, which were associated with adaptation to high salinity. While more roots were maintained in their primordial state under SST, the root surface area was significantly increased, achieving enhanced metabolite uptake and active ion efflux and improved root-rhizosphere interaction. Transcriptional changes also indicate increased deposition of lignin in cell walls, which likely serves to create an apoplastic barrier to prevent water and solute loss and to reduce ionic flow through the apoplastic pathway [64]. High rate of Na^+^ accumulation in the tissues of SST treated plants may be caused by Na^+^ replacement of Ca^2+^ in cell walls [65,66,67]. Even though Na^+^ was highly accumulated in SST-treated tissues, its concentration in the shoots was 1.4-fold lower than in the roots, implicating active Na^+^ recruitment as an additional cheap osmotic agent in the latter, and suggesting prevention of Na^+^ transport to photosynthetic tissues. It is worth mentioning that the experimental studies demonstrate much higher level of Na^+^ accumulation in shoot tissues of *H. vulgare* [16,18,20]. Accordingly, few changes at the metabolic and transcriptional levels were detected in SST-treated shoots, when compared with the roots (Table 2). Changes in transcription levels of genes encoding transport proteins were particularly extensive in the roots (Table 2) and are considered to correspond to prevention/deceleration of toxic ion accumulation and assurance of nourishment and water uptake. Despite K^+^ being generally regarded as a main competitor of Na^+^ in uptake and transport [5], K^+^ content was decreased in stressed roots, but the K^+^/Na^+^ ratio was higher in the shoots. The K^+^ decrease is likely caused by stress-induced K^+^ leakage and competition with Na^+^ in root environment [5]. The K^+^/Na^+^ ratio of 1.4 achieved in our experiments is very similar to that observed in other *H. marinum* ecotypes and differs strongly from the K^+^/Na^+^ ratio of 4.3 found in *H. vulgare* plants [20]. These results further support the idea that the salt tolerance of *H. marinum* may be based on maintaining Na^+^/K^+^ balance in its shoots under salinity [16,18,20,59,68]. Despite the proposed roles of SOS1, HKT1;1, HKT1;5, and HKT2;2 in establishing this balance in *H. marinum* under salinity [18], none of the corresponding genes were found to be differentially expressed in our study. A significant decrease of *HmHKT2;1* transcript was observed in one *H. marinum* ecotype but remained unchanged in plants of another ecotype after SST [20]. Instead, other transporters, including K^+^ transporter SKOR, an ammonium transporter, a cation/H^+^ antiporter, and a Mg^2+^ transporter, were strongly transcriptionally upregulated in SST-treated roots, and may therefore enhance cation uptake and xylem loading to compete with Na^+^. Decreased expression of *GLR3.4* and a *cyclic nucleotide-gated channel*, both potentially nonspecific Na^+^ channels, may also contribute to minimization of Na^+^ uptake by the roots. In SST-treated *H. vulgare* roots, expression of *HvHKT1;5* and *HvSOS1* was also decreased, whereas that of *HvSKOR* was increased [69]. Besides substitution of Na^+^ by K^+^ in the cytosol, Na^+^ compartmentalization into the vacuoles or endosomes may serve as an additional mechanism of salinity tolerance in *H. marinum*. However, this vacuolar or endosomal sequestration could not be explained by the expression of NHX transporters and thus remains unclear. While Ca^2+^ has been described as an early component of salt sensing [5], the Ca^2+^ level under SST was decreased, likely due to its replacement by Na^+^ in cell walls and vacuoles, the compartments with the largest Ca^2+^ pools in plants [65,66,67]. It would be a good option to study functions of Na^+^ and K^+^ transport proteins, in particular HmSKOR, with further prospective application in domesticated cereals.

The roots of *H. marinum* are also likely to be able to control Cl^−^ ions under high salinity. The increased expression of S-type anion channels *SLAH2* and *SLAH3* specifically under SST is probably connected with Cl^−^ retrieval from the xylem and/or efflux from the roots. The repression of *MSL10*, a channel with a moderate Cl^−^ preference [70], may further reflect the reduction of Cl^−^ uptake. Upregulation of four boron transporters in SST-treated roots is notable. These transporters belong to the anion exchanger family [71], do not have strict boron selectivity and may transfer other anions including Cl^−^, contributing to its removal. Combined salt and boron tolerance have been frequently described [72,73]. Increased expression of several *S* and *P transporter* genes, as well as *NRT1/PTR family protein* genes, may indicate activation of anion uptake to compete with Cl^−^. It would be of interest to test the role of these transporters in salinity tolerance and their suitability for biotechnological improvement of other crops.

In OST-treated plants, a different group of specific cation and anion transporters was upregulated, whose activity may result in the increased cation (Mg^2+^, Ca^2+^, Zn^2+^) and anion (MoO_4_^2−^, SO_4_^2−^, PO_4_^3−^) uptake required for osmotic adjustments. Transcriptional depletion of key salinity tolerance genes *HKT1;5* and *HKT1;4*, along with upregulation of *Na^+^/H^+^ exchanger 5* in the shoots, implies endosomal sequestration of Na^+^/K^+^ for cellular osmotic rearrangements. Marked upregulation of different Zn transporters was specifically detected in OST-treated roots, in line with increased Zn^2+^ accumulation in the shoots. Zn^2+^ has been shown to be involved in ROS scavenging [74] and stomata opening via determination of the K^+^ influx rate [75]. Increased Zn content in the shoots under OST is thought to facilitate ROS detoxification and regulation of water management.

Strong transcriptional upregulation of amino acid-degrading enzymes in the roots under both types of stress was coupled with upregulation of genes encoding enzymes involved in utilizing degradation products to produce secondary metabolites. The accumulation of flavonoids occurred specifically in SST-treated roots, whereas their biosynthesis was likely repressed under OST. Accumulation of isoprenoids and carotenoids might be also repressed under OST. While flavonoids may function as ROS scavengers, some (e.g., chalconoids) are able to block voltage-dependent K^+^ channels [76]. Ectopic expression of chalcone synthase has been shown to increase salt tolerance [77]. Increased flavonoid accumulation in *H. marinum* is thought to contribute to the inhibition of stress-induced K^+^ leakage and ROS balance. Due to transcriptional upregulation of numerous tyrosine decarboxylase genes, involved in the production of little-studied phenylethylamine hordenine [78], the role of alkaloid hordenine in stress tolerance has become obvious and deserves further investigation.

To conclude, the mechanisms of salt tolerance of seaside barley are complex and comprised of adaptations on morphological, physiological, biochemical, and transcriptomic levels (Figure 7C). Seaside barley is likely capable of controlling Na^+^ and Cl^−^ concentrations in its leaves when its roots are subjected to high salinity. However, transporters, shown to achieve salinity tolerance [14,79], were unchanged in the plant line used in the present study, in line with variability of *H. marinum* accessions in salinity tolerance [80,81]. The adaptation of *H. marinum* to SST includes fine-tuning of membrane transport for effective management of both restriction of ion entry and sequestration, as well as accumulation of osmolytes, which help to minimize energy costs. In contrast, markedly more resources and energy are required to overcome the negative consequences of OST, particularly due to the severity of ROS accumulation and nutritional imbalance affecting plant growth under stress. Our results demonstrate that, in order to adapt to salinity, seaside barley has developed specific mechanisms that differ from those which are activated in response to OST.

## 4. Materials and Methods

### 4.1. Plant Material and Growth Conditions

Seeds of seaside barley (*Hordeum marinum* ssp. *marinum*), originated from Tuscany region, and cultivated barley (*Hordeum vulgare*) cv. Golden Promise were germinated on moist filter paper in the dark at 20 °C. Seven day-old seedlings were transferred on hydroponic half-strength Hoagland’s No. 2 solution (Sigma-Aldrich, St. Louis, MO, USA) and incubated in the growth chamber under irradiance of 350 mmol m^−2^ s^−1^, 12 h photoperiod, and 20 °C. Due to the halophytic nature of *H. marinum*, 0.2 mM NaCl was added to the control incubation solution [15]. The incubation solution was fully replaced every week by a newly prepared one in order to prevent nutrient depletion. After 14 days of growing, plants were exposed stepwise to the increasing concentrations of 50 mM NaCl or 2.5% PEG6000 per day until 300 mM NaCl (salinity stress) or 15% PEG6000 (osmotic stress) were reached. Control plants were exposed to 0.2 mM NaCl continuously. All plants were sampled after five days (32 days old) of maximum stress and separated into shoot (crown and growing point) and root (2 cm root tips) fractions. To average the genetic background and local environmental influences, a bulk of 10–15 plants, grown in a single hydroponic tank either under the control or stress conditions, were collected for one biological replication. In total, 10 biological replications were harvested, frozen, and used in further analyses.

### 4.2. Determination of Morphological and Physiological Characteristics

Fresh plant tissues were collected in 1.5 mL Eppendorf microcentrifuge tubes. The tissue sap was excavated by squeezing whole plants with Pellet pestle (Eppendorf, Germany). Osmotic pressure of experimental solutions and sap obtained from squeezed whole plants was measured by Vapor Pressure Osmometer WESCOR 5500 (Thermo Fisher Scientific GmbH, Dreieich, Germany) according to manufacturer’s instructions. The relative growth rate (RGR) was calculated from the fresh weight data taken at start of stress application and final harvest using the formula RGR = (ln fresh weight_2_ − ln fresh weight_1_)/(t_2_ − t_1_), where fresh weight_1_ = fresh weight (g) at t_1_; fresh weight_2_ = fresh weight (g) at t_2_; and t_1_ and t_2_ = time at start and end of experiments in days. The RGRs of individual plants were presented as percentages relative to value in control conditions. Plant water content (PWC) in *H. marinum* and *H. vulgare* plants was determined as follows: PWC = (FW − DW)/DW, where DW was dry weight and FW was fresh weight of an individual plant. The PWC values were converted into percentages relative to the value in the control condition. To estimate shoot/root ratio in tested plants, the FW of shoots and roots of individual plants of *H. marinum* and *H. vulgare* were measured after end of stress application.

### 4.3. Elemental Analysis

Approximately 10 mg of pulverized and dried (at 65 °C) plant material was weighed into PTFE digestion tubes and 1 mL of concentrated nitric acid (67–69%) was added to each tube. After 4 h incubation, samples were digested under pressure using a high-performance microwave reactor Ultraclave 4 (MLS, Leutkirch, Germany). Samples were then transferred to Greiner centrifuge tubes and diluted with de-ionized water to a final volume of 8 mL. Elemental analysis was carried out using a sector field high resolution mass spectrometer (HR)-ICP-MS ELEMENT 2 (Thermo Fisher Scientific, Dreieich, Germany) with Software version 3.1.2.242. A 10 points external standard calibration curve was set from a certified multiple standards solution (Bernd Kraft, Germany). A least-square regression was applied to best fit the linearity of the curve. Elements Rh and Ge (ICP Standard Certipur^®^, Merck, Germany) were infused online and used as internal standards for matrix correction.

### 4.4. Non-Invasive NMR-Imaging and NMR-Spectroscopy of Plant Tissues

The NMR imaging of *H. marinum* tissue was conducted according to [82]. In total, three plants for each type of treatment were analyzed. The internal tissue structure of stressed and control plants was visualized noninvasively with an isotropic resolution of around 40 µm. The NMR analysis was conducted on a Bruker Ascend TD 400 MHz NMR spectrometer (Bruker GmbH, Rheinstetten, Germany). Image processing was performed by application of software MATLAB (The Mathworks, Natick, MA, USA) and AMIRA (Thermo Fisher Scientific GmbH, Dreieich, Germany).

### 4.5. Measurement of ^13^C Uptake

400 ppm ^13^CO_2_ was applied for 24 h to bag-covered hydroponic tanks with control or stressed plants grown in the chamber in the above-described conditions. Afterwards, treated plants were separated into shoot and root fractions, lyophilized, ground, and analyzed on elemental analyzer coupled to stable isotope ratio mass spectrometer (Vario MICRO cube/Isoprime Vision, Elementar Analysensysteme GmbH, Langenselbold, Germany). Five biological repetitions with three technical replicates each were analyzed.

### 4.6. Untargeted Metabolite Profiling

For the untargeted analysis of central metabolites, the freeze-dried and homogenized samples were incubated for 20 min at 4 °C in 600 µL of extraction buffer consisting of equal volumes of methanol and chloroform (Roth, Karlsruhe, Germany) followed by addition of 300 µL of water and centrifugation at 14,000 rpm at 4 °C for 10 min. Supernatant was transferred into a new tube and stored at −80 °C, prior to the analysis by ion chromatography using Dionex-ICS-5000+HPIC system (Thermo Scientific, Dreieich, Germany) coupled to a Q-Exactive Plus hybrid quadrupol-orbitrap mass spectrometer (Thermo Scientific, Dreieich, Germany). The detailed chromatographic and mass spectrometry (MS) conditions are described in the Appendix A. The randomized samples were analyzed in full MS mode. The data-dependent MS-MS analysis for the compound identification was performed in the pooled probe, which also served as a quality control (QC). The batch data was processed using the untargeted metabolomics workflow of the Compound Discoverer 3.0 software (Thermo Fisher Scientific, Dreieich, Germany). The compounds with the maximum relative standard deviation (RSD) below 35% of the QC area were selected for quantification. The compounds were identified using the inhouse library, as well as a public spectral database mzCloud, and the public databases KEGG, NIST and ChEBI via the mass- or formula-based search algorithm. The *p*-values of the group ratio were calculated by ANOVA and a Tukey-HCD post hoc analysis. Adjusted *p*-values were calculated using Benjamini-Hochberg correction.

### 4.7. RNA Extraction, Sequencing and Transcript Analysis

The total RNA was isolated using TRIzol (Thermo Fisher Scientific, Schwerte, Germany) with subsequent DNAse treatment (Thermo Fisher Scientific, Schwerte, Germany) and additional purification using Plant RNA purification kit (Qiagen, Hilden, Germany). cDNA libraries were prepared using Lexogen SENSE RNA-Seq Kit (Lexogen, Vienna, Austria) and sequencing was performed in HiSeq2500 device (Illumina, San Diego, CA, USA). Three repetitions were performed for each data point. The complete data set is deposited at the European Nucleotide Archive with accession number: PRJEB38377.

Adapter trimming was performed using Cutadapt software, version 1.9.1 [83]. Quality trimming was performed using CLC assembly cell software, version 5.0.1 (Qiagen, Hilden, Germany). Read mapping was performed on the *H. vulgare* genome [48] using the software Kallisto, version 0.45.0 [84]. Differential expression was calculated using the R package DESeq2, version 1.18.1 [85]. Differential expression thresholds were set at log_2_-fold change > 1.5 and FDR-adjusted *p* values (according to Benjamini Hochberg) < 0.01. Venn diagrams were created using InteractiVenn [86]. Gene ontology (GO) terms enrichment analysis for biological process carried out using the GO enrichment tool by Panther [87].

### 4.8. Statistical Analysis

Significance analysis was performed by Student’s *t* test using BBBB software (IBM SPSS Statistics, version 16). The difference at *p* < 0.05 was considered as significant. For data presentation, significance was marked as: *, *p* < 0.05; **, *p* < 0.01; ***, *p* < 0.001.

## Figures and Tables

**Figure 1 ijms-21-09019-f001:**
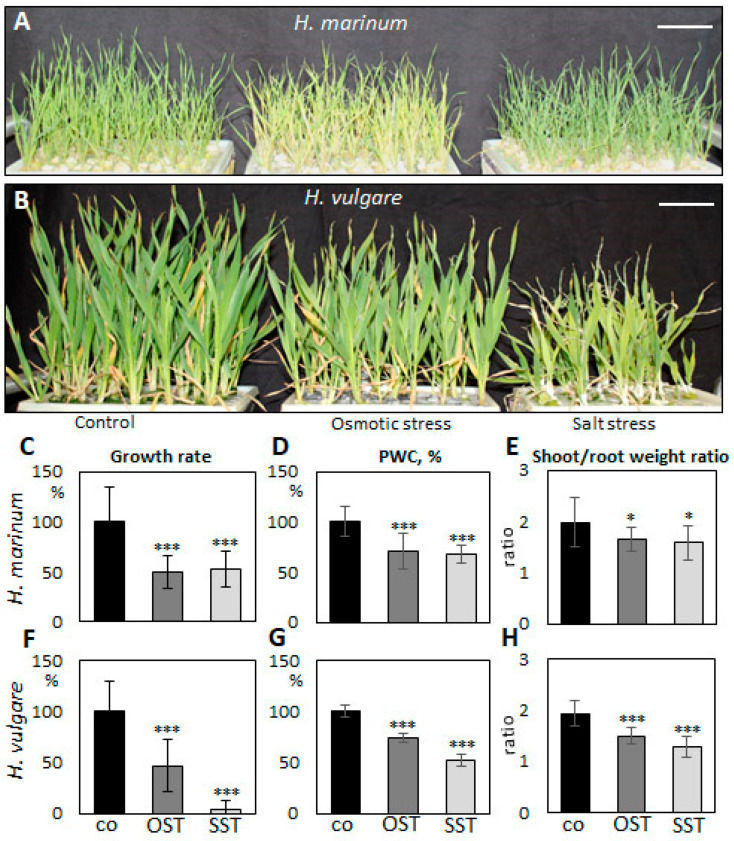
Changes in growth of *Hordeum marinum* and *H. vulgare* plants under osmotic (OST) and salinity (SST) stresses. (**A**,**B**) Morphological characteristics of *H. marinum* (**A**) and *H. vulgare* plants (**B**) under SST and OST after reaching the maximum stress (27 days old); (**C**,**F**) relative growth rate, (**D**,**G**) plant water content (PWC), and (**E**,**H**) shoot/root weight ratio of *H. marinum* (**C**–**E**) and *H. vulgare* (**F**–**H**) plants under control and stressed conditions. Scale bars = 5 cm. Data are mean ± SD; *n* = 8, *t* significant at: *, *p* < 0.05, and ***, *p* < 0.001.

**Figure 2 ijms-21-09019-f002:**
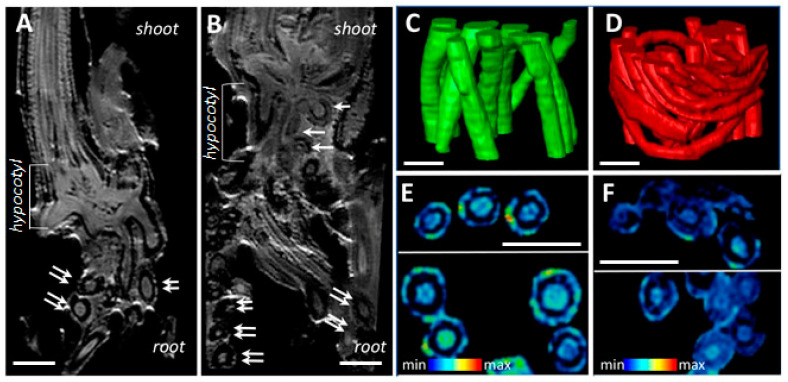
Comparative non-invasive magnetic resonance imaging (MRI) revealed structural changes in SST roots of *H. marinum* when compared to the control. (**A**,**B**) The representative virtual cross-sections show the internal structure of hypocotyl regions of plants growing under the control condition (**A**) and SST (**B**). Numerous root nodules in the hypocotyl region (white arrows) and root cross-sections (doubled arrows) are visible. (**C**,**D**) Fragments of the 3D models show spatial arrangement of the fibrous roots in control (**C**, green) and SST (**D**, red). (**E**,**F**) Relative differences in water distribution across the root tissues are visualized in virtual cross-sections of control (**E**) and SST (**F**) roots. MRI signal in (**E**,**F**) is at an identical scale and represented using a rainbow-based color scheme. High signal intensities in red (max) indicate high water saturation, while the blue regions (min) indicate lower water saturation. Scale bars = 1 mm.

**Figure 3 ijms-21-09019-f003:**
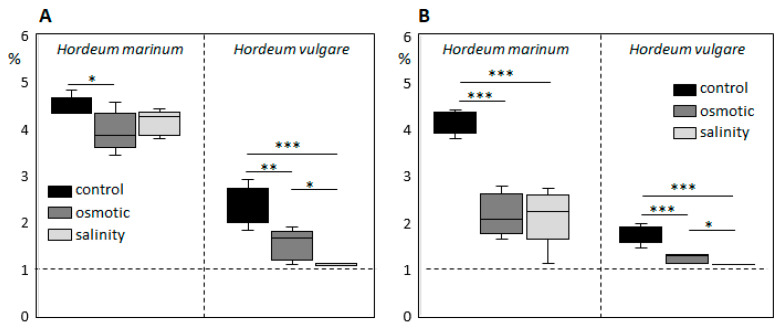
^13^C uptake and distribution in shoots (**A**) and roots (**B**) of *Hordeum marinum* and *H. vulgare* plants under control conditions as well as under osmotic and salinity stresses. Dashed lines indicate natural ^13^C abundance. Data are mean ± SD; *n* = 5, *t* significant at: *, *p* < 0.05; **, *p* < 0.01; ***, *p* < 0.001.

**Figure 4 ijms-21-09019-f004:**
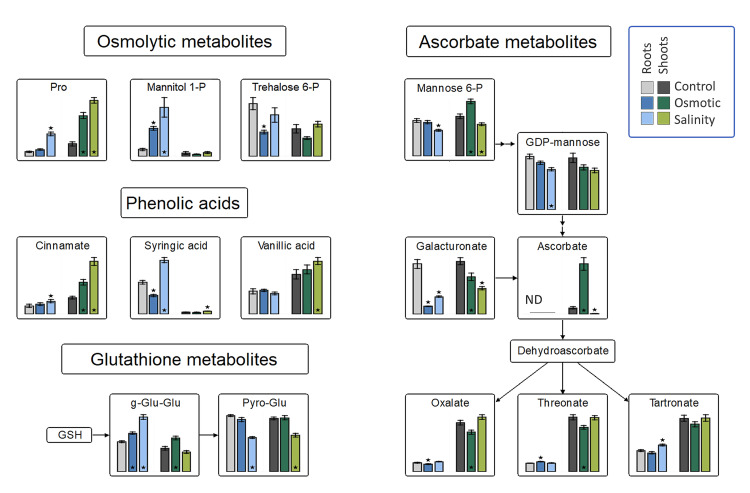
Changes in osmolytic metabolites, and the antioxidant system in roots and shoots of *Hordeum marinum* under osmotic and salinity stresses. Bars represent means of seven independent replicates ± SE. Significant differences to control treatments at specified time points after excision are indicated by asterisks (Wilcoxon, Mann-Whitney U-test; *, *p* < 0.05).

**Figure 5 ijms-21-09019-f005:**
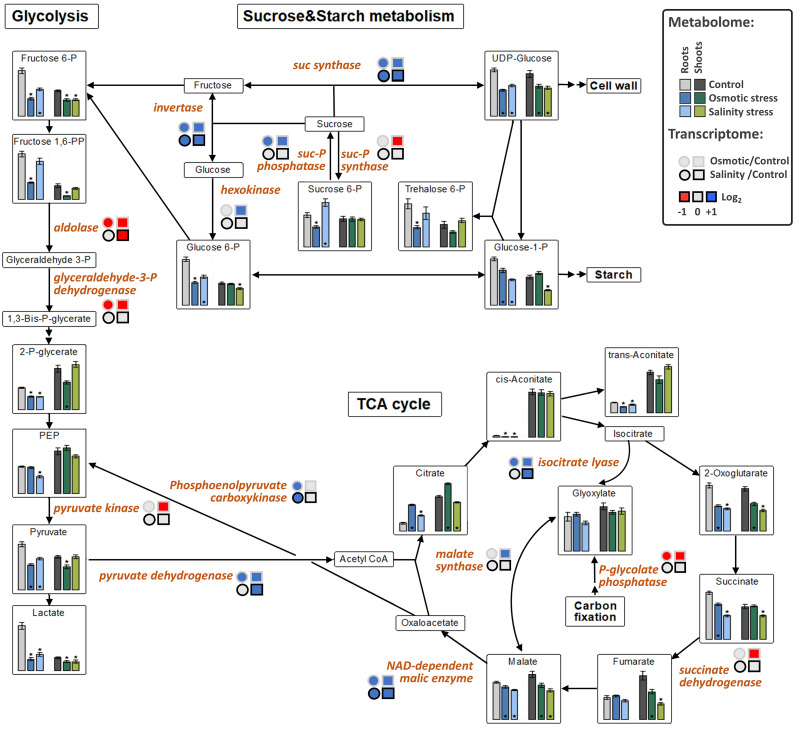
The effect of osmotic and salinity stresses on the sugar and central metabolism and corresponding transcriptomic changes in roots and shoots of *Hordeum marinum.* Metabolic data, presented as bars, are means ± SE; *n* = 7. Significant differences to control treatments at specified time points after excision are indicated by *, *p* < 0.05 (Wilcoxon, Mann-Whitney U-test). Up-regulated genes are labeled by blue, down-regulated by red color.

**Figure 6 ijms-21-09019-f006:**
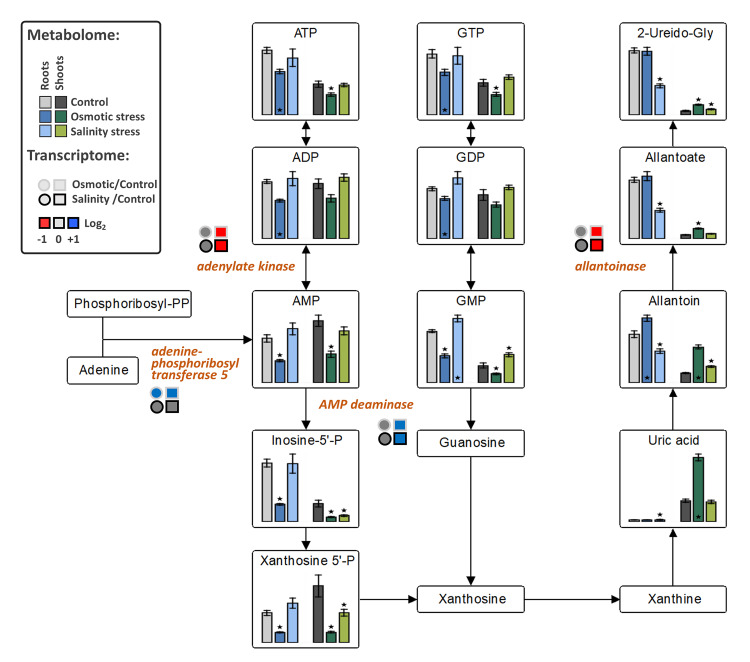
The effect of osmotic and salinity stresses on the purine catabolism and corresponding transcriptomic changes in roots and shoots of *Hordeum marinum.* Metabolic data, presented as bars, are means ± SE; *n* = 7. Significant differences to control treatments at specified time points after excision are indicated by *, *p* < 0.05 (Wilcoxon, Mann-Whitney U-test). Up-regulated genes are labeled by blue, down-regulated by red color, and those with no changes in expression by grey color.

**Figure 7 ijms-21-09019-f007:**
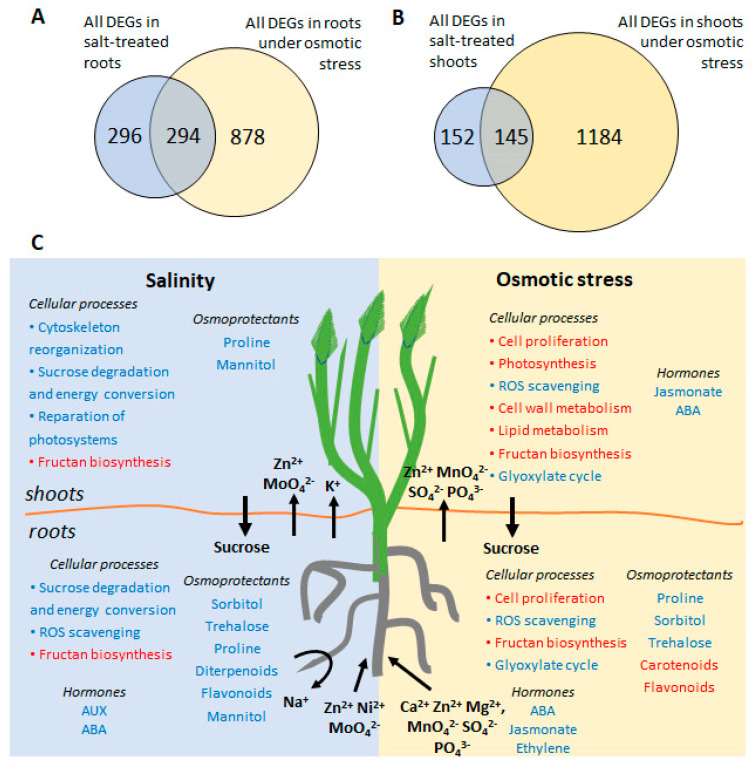
(**A**,**B**) Venn diagram showing the numbers of common and stress-specific differentially expressed genes (DEG) in roots (**A**) and shoots (**B**) after osmotic (15% PEG6000) and salt (300 mM NaCl) stresses. **(C)** A schematic overview of the main processes occurred during salinity and osmotic stress in *H. marinum*. Activated processes are highlighted in blue, inhibited processes are highlighted in red. Arrows indicate directions of mineral, phytohormone, and sugar redistribution between roots and shoots upon stress treatment.

**Table 1 ijms-21-09019-t001:** Element compositions in shoots and roots of *Hordeum marinum* under osmotic and salinity stress compared to control.

Element	Shoots, (µg/g) DW *			Roots (µg/g) DW *		
	Control	Osmotic Stress	Salinity	Control	Osmotic Stress	Salinity
**^11^B**	17.9 ± 3.9	13.7 ± 3.1 *	13.3 ± 0.7 **	5.8 ± 1.2	5.7 ± 0.7	5.0 ± 0.8
**^98^Mo**	1.8 ± 0.1	2.4 ± 0.5 **	3.4 ± 0.3 ***	1.8 ± 0.9	1.2 ± 0.1	2.8 ± 0.1 **
**^31^P**	6510.8 ± 276.4	8006.4 ± 371.0 ***	5696.5 ± 191.9 ***	7152.7 ± 167.5	7850.4 ± 252.0 ***	6463.2 ± 199.8 ***
**^44^Ca**	6958.7 ± 1596.1	6364.0 ± 394.0	2604.8 ± 424.6 ***	2806.7 ± 375.5	6850.9 ± 752.6 ***	1388.8 ± 282.9 ***
**^55^Mn**	108.1 ± 14.8	205.7 ± 49.2 ***	91.1 ± 37.6	199.4 ± 27.0	379.4 ± 36.6 ***	200.1 ± 23.3
**^60^Ni**	4.5 ± 2.3	9.8 ± 3.7 **	6.5 ± 5.8	6.5 ± 3.2	13.8 ± 3.2 ***	16.7 ± 8.2 **
**^63^Cu**	20.5 ± 2.1	38.7 ± 48.3	19.8 ± 4.7	297.9 ± 18.0	183.3 ± 13.9 ***	282.8 ± 20.3
**^66^Zn**	35.3 ± 2.3	74.1 ± 6.3 ***	56.4 ± 4.8 ***	44.2 ± 5.1	35.7 ± 2.1 ***	111.1 ± 7.0 ***
**^23^Na**	539.2 ± 162.3	545.9 ± 68.7	25614.5 ± 2381.9 ***	728.2 ± 81.8	505.7 ± 51.6 ***	36680.6 ± 3394.2 ***
**^26^Mg**	3048.6 ± 680.1	2805.7 ± 415.2	1504.1 ± 201.5 ***	1391.7 ± 233.3	1730.4 ± 51.0 **	1075.7 ± 83.3 **
**^34^S**	449439 ± 353.8	5845.4 ± 942.7 **	3329.9 ± 249.6 ***	3125.5 ± 191.7	3544.4 ± 162.8 ***	3123.6 ± 155.7
**^39^K**	54968.5 ± 3184.1	52948.5 ± 6022.5	42608.6 ± 2273.7 ***	41786.5 ± 1919.9	37967.2 ± 2353.3 **	27505.5 ± 1466.2 ***

* DW, dry weight. Significantly increased contents are highlighted in blue, significantly decreased contents are highlighted in red. Data are means ± SD, n = 8–10, * *t* significant at *p* < 0.05, ** *t* significant at *p* < 0.01 and *** *t* significant at *p* < 0.001.

**Table 2 ijms-21-09019-t002:** Gene ontology (GO) term enrichment in the differentially expressed genes (DEGs) from the specific tissue under osmotic or salt stresses.

Biological Process	Fold Enrichment in
OST Roots	OST Shoots	SST Roots	SST Shoots
Tricarboxylic acid metabolism (GO:0072351 + GO:0072350)	9.56–16.73	7.84–13.55		
Generation of precursor metabolites and energy (GO:0006091)		2.06		
Nicotianamine metabolism (GO:0030418 + GO:0030417)	16.73	13.55		
Amine metabolism (GO:0009309 + GO:0044106)	5.01–5.52			
Cold acclimation and response to cold (GO:0009631 + GO:0009409)	14.6		6.91–29.86	
Nitrate response and transport (GO:0010167 + GO:0015706)	9.61–10.14			
Transition metal ion transport (GO:0000041)	4.27			
Anion transport (GO:0015698 + GO:0006820 + GO:0098656)	2.79–4.25		3.51–5.62	
Ion transport (GO:0006811)	1.97		2.37	
Transmembrane transport (GO:0055085)	1.85		2.06	
Response to inorganic substances (GO:0010035)	3.21			
Response to acid chemical (GO:0001101)	2.85	2.8	3.67	
Oxidation-reduction (GO:0055114 + GO:0072593 + GO:0098869)	1.71	1.84–3.24	2.1	2.13
Tryptophan metabolism (GO:0000162 + GO:0006568)			11.87–15.37	
Indole compound metabolism (GO:0042435 + GO:0042430)			11.87–15.37	
Indolalkylamine metabolism (GO:0046219 + GO:0006586)			11.87–15.37	
Response to abscisic acid (GO:0009737)			5.74	
Response to alcohol (GO:0097305)			5.68	
Response to lipid (GO:0033993)			4.11	
Drug metabolism (GO:0042737+ GO:0017144)		2.54–2.8	3.33	
Photosynthesis (GO:0015979 + GO:0009768 + GO:0009765)		3.88–8.55		
Chromatin organization (GO:0097549 + GO:0045814 + GO:0034401)		5.2–5.41		
Antibiotic metabolism (GO:0016999 + GO:0017001)		3.02–3.14		
Cofactor metabolism (GO:0051187 + GO:0051186)		2.07–3.01		
Cellular detoxification (GO:1990748 + GO:0097237)		2.5		
Small molecule biosynthetic process (GO:0044283)		2.01

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
