# Peer review of "Adaptation Strategies of Halophytic Barley Hordeum marinum ssp. marinum to High Salinity and Osmotic Stress"

_ijms, 2020, doi:10.3390/ijms21239019_

Round 1

Reviewer 1 Report

The manuscript by Isayenkov et al., is of great interest for the adaptation of halophytic barley to high salinity. The experiments were elaborately planned. The results were well described. I have minor suggestions that can strengthen the manuscript.

Figure 1C-K are not presented well. It would help the readers understand the findings better if these figures were organized differently. Figure 1I-K looked it they were a part of the rest of the figure. These should be separated and labeled clearly.

Line 84 “whereas the H. marinum plants maintained their strong 84 green color (Figure 1A, B)”. This should be accompanied with chlorophyll contents for these plants.

Differences in phenotypes of control and salt stress plants in Fig1B don’t really show that plants under salt stress exhibited 95% of growth reduction as shown in 1F. Perhaps the authors should select images that represent this difference.

The authors reported comprehensive transcript profiling under OST and SST. Have the authors used knockout mutants or overexpression of genes found to be regulated under OST and SST to determine if they contribute to any phenotypic changes observed under these conditions?

The authors showed that only minor transcriptional changes in genes involved in general cellular processes and in the metabolism of lipids and the cell were observed. Aside from that, lipids paly important roles in permeability of cell membrane which might be important for salt resistance. Have the authors analyzed fatty acid compositions in plants under SST?

Author Response

Respond to comments of the reviewer 1

Dear Reviewer,

Thank you very much for your thorough reading of the manuscript and critical evaluation. We have submitted new version of our manuscript where you can find all corrections made according to your suggestions and remarks. Our respond to your comments is given below.

The manuscript by Isayenkov et al., is of great interest for the adaptation of halophytic barley to high salinity. The experiments were elaborately planned. The results were well described. I have minor suggestions that can strengthen the manuscript.

A: We appreciate your comment.

Figure 1C-K are not presented well. It would help the readers understand the findings better if these figures were organized differently. Figure 1I-K looked it they were a part of the rest of the figure. These should be separated and labeled clearly.

A: We agree that the Fig. 1 was suboptimal. We separated and transferred the Fig.1I-K to supplementary material as Fig. S1.

Line 84 “whereas the H. marinum plants maintained their strong 84 green color (Figure 1A, B)”. This should be accompanied with chlorophyll contents for these plants.

A: Unfortunately, we did not measure chlorophyll content in plants. The primary aim of our study was to compare transcriptomic and metabolite responses to salinity and osmotic stress. We will consider your advice for future studies with this halophytic barley. In this work, the data of 13C assimilation and transcript and metabolite profiling strongly suggest that efficiency of photosynthesis is reduced due to osmotic stress (line 267 and lines 431 – 434) that correspond to the observed phenotype of yellowish leaves.

Differences in phenotypes of control and salt stress plants in Fig1B don’t really show that plants under salt stress exhibited 95% of growth reduction as shown in 1F. Perhaps the authors should select images that represent this difference.

A: The Figure 1A and B show the plants after reaching the maximum stress (or 27 days old) and not at the end of the experiment (32 days). Maximal stress application for additional five days (when the plants were sampled) disastrously affected especially H. vulgare growth (see attached figure below). Therefore, we decided to leave the original image, because it shows clear phenotypic differences of two barley species in the middle of the stress treatment. We added to the legend the exact time when the images were taken.

The authors reported comprehensive transcript profiling under OST and SST. Have the authors used knockout mutants or overexpression of genes found to be regulated under OST and SST to determine if they contribute to any phenotypic changes observed under these conditions?

A: In our experiments, we have used wild halophytic barley species for which no mutations is known. Besides, this species is still not able to be transformed. However, we consider to clone some interesting genes to conduct the functional analysis and express them in heterologous systems (including H. vulgare) in future projects.

The authors showed that only minor transcriptional changes in genes involved in general cellular processes and in the metabolism of lipids and the cell were observed. Aside from that, lipids paly important roles in permeability of cell membrane which might be important for salt resistance. Have the authors analyzed fatty acid compositions in plants under SST?

A: Unfortunately, we did not analyse the fatty acid composition in our study. We will plan this work in future study mentioned above.

Reviewer 2 Report

The manuscript presents a thorough analysis of responses of H. Marinum to salt stress. The richness of analytical methods used allows to gain possibly new insights on tolerance mechanisms. I have some concerns about the comparability of these results to the response of H. marinum to of PEG-caused osmotic stress, and the comparison with H. vulgare, which makes a too separate compartment in the manuscript. H. vulgare is included in the research, for comparative purposes, and because barley is an important crop, but this comparison is not carried very far. Most of the work is performed on H. marinum, as acknowledged in the title of the manuscript. The manuscript would have given more useful insights if the comparison with H. vulgare (the cultivated relative), which was limited to morphological and a few physiological measurements, had been extended to NMR, metabolomics and gene expression analyses. Further analyses on the vulgare samples would have provided better insights about the similarities and differences of responses, with implications for plant breeding. Are these analyses still possible? I would appreciate comments on how the mechanisms found in marinum differ from those found in vulgare (in this or in other studies), and if and how mechanisms found in marinum could be imported into vulgare.

Authors should acknowledge that salinity stress is actually a complex stress with osmotic and toxic components, which usually predominate in different parts of the plant’s growth cycle. Osmotic is prevalent while plants are producing new tissues, because of the diluting effect of incoming salts. When leaf growth is halted in annual plants, and grain growth dominates, a toxic effect of salt is prevalent. Different mechanisms act under these two stages of salinity stress. Which is the stage of the plants when they are sampled? Are they still producing new leaves and, therefore, still taking advantage of the dilution effect? The authors should put their findings in the context of the stage of plant growth, and comment on the prevalence of osmotic and toxic stresses in their SST treatment. Could you please comment on this?

Classically, osmotic stress due to drought and salinity stress have been compared at similar osmotic stress levels in the root media, caused by different osmotic agents. This seems not the case here (Fig. 1 I), as osmotic stress caused by salinity was much higher than caused by PEG. Can the authors comment on why were the stress levels chosen? Please, justify the validity of comparisons between two stresses which induce so different osmotic stress to the plant. Is it possible to predict what would an osmotic stress caused by PEG, of the same intensity as the one caused by salt, have done? The authors should provide a convincing reply to this concern, to justify their choice, because the comparison of the two types of stress is the backbone of the menuscript.  

It is not clear what new findings are contributed by this study. What is reported reminds of rather similar findings in previous studies. The authors should make an effort to highlight their novel findings, particularly in the intersection of metabolome-transcriptome-physiology, which is the strength of the study.

Several aspects of methodology and experimental design need clarification:

L543-544. “The incubation solution was replaced by fresh weekly in order to prevent nutrient depletion” This means that the solution was replaced fully every week? Please clarify the sentence

L548-549. “10-15 plants per hydroponic tank, representing a single replicate of treatment, were harvested, frozen and used in further analyses.” The use of the term “replicate” here is misleading. Was a single replicate used? In Figure 1, n=8, which I understand as 8 biological replicates (probably single plants). Apparently, the number of biological replicates was 8, not 1. Please give a full description of the experimental set-up and experimental design, and clarify the replicates issue, and the nature of the experimental unit (one plant or a bulk of several plants?) Please indicate in the M&M section the number of biological replicates used for each assay: NMR, untargeted metabolite profiling (n=7), mineral analysis (n=8-10), C13 (n=5). State clearly the number of biological reps for RNA-seq: “Three repetitions were performed for each data point”, were this technical or biological?

Also, where all plants of H. marinum and H. vulgare grown in single tanks? Or was there any kind of blocking structure?

How many harvests were actually done? L547, the text describes a single harvest, at day 32. But then calculation of RGR (L555) requires at least two harvests, at the beginning and end of the experiment.

The method to measure leaf osmotic pressure is not explained. This is absolutely needed. The equipment used is not specific for plant tissues. Specific protocols for plant tissues (leaf discs, extracted sap, etc) have been developed (Ball and Oosterhuis, 2005, Env Exp Bot), and used in plant research (for instance Caringella et al 2015, PCE). The authors should describe with enough detail the procedures used, the tissues sampled, etc. Also, I recommend expressing osmotic pressure in the more usual MPa units, to facilitate comparison with other studies.

Other issues:

L89 (Figure 1). Not clear what panels I-J-K are

I miss measurement of leaf potential in the report. Was it possible to measure leaf potential in the samples? Please, comment on this matter.

Is PWC the same as relative water content, which is a trait more commonly found in the literature?

In Figure 3, what catches the eye is the much higher uptake of C13 by marimun, compared to vulgare, even at the control treatment. Was this expected? Is there an explanation for this?

L142. Table 1. Please align numbers in the table. Does colour coding mean anything? In the footnote, include “*t significantly different from the control treatment at…”

L174, please use the term “Principal component analysis”

L276. Please, align Table (maybe landscape?)

L617. “The P values were adjusted for multiple testing according  to Benjamini-Hochberg”. This statement is void unless it is accompanied by more information, like the actual q-value used as cut-off.

Evidences of restriction of ion entry and sequestration are not that big (L530).

Author Response

Respond to the comments of the Reviewer 2

Dear Reviewer,

We appreciate your comments and constructive suggestions. Our specific (point by point) reply to your comments is given below.

The manuscript presents a thorough analysis of responses of H. Marinum to salt stress. The richness of analytical methods used allows to gain possibly new insights on tolerance mechanisms. I have some concerns about the comparability of these results to the response of H. marinum to of PEG-caused osmotic stress, and the comparison with H. vulgare, which makes a too separate compartment in the manuscript. H. vulgare is included in the research, for comparative purposes, and because barley is an important crop, but this comparison is not carried very far. Most of the work is performed on H. marinum, as acknowledged in the title of the manuscript. The manuscript would have given more useful insights if the comparison with H. vulgare (the cultivated relative), which was limited to morphological and a few physiological measurements, had been extended to NMR, metabolomics and gene expression analyses. Further analyses on the vulgare samples would have provided better insights about the similarities and differences of responses, with implications for plant breeding. Are these analyses still possible? I would appreciate comments on how the mechanisms found in marinum differ from those found in vulgare (in this or in other studies), and if and how mechanisms found in marinum could be imported into vulgare.

  1. In our work, we focused our research on H. marinum, a wild halophytic relative of the cultivated H. vulgare. There is plenty of metabolic and transcriptomic research regarding responses of different H. vulgare cultivars to salinity and osmotic stress (e.g., recently published Osthoff et al. 2019, Yousefirad et al. 2020, Hill et al. 2016, and many others). Although H. vulgare is regarded as relatively salt tolerant when compared with other cultivated Triticeae, cultivated barley experiences still a 55–58% decline in biomass at less than one-third the NaCl concentration in sea water (Colmer et al., 2006). The wild sea-side barley (H. marinum) is capable to grow and reproduce in condition of salt marshes and coastal areas with eventual sea water logging (Garthwaite et al., 2005). H. vulgare belongs to includer type of plants and can accumulate substantial amount of Na in tissues (Mian et al., 2011). The adaptation mechanism of H. marinum to high salinity is expected to be different due to its halophytic nature. The evaluation of these mechanism was a main goal of a study. We intend to transfer the knowledge received in this investigation to H. vulgare by overexpression of some key transporter genes to reach enhanced salinity tolerance.

Authors should acknowledge that salinity stress is actually a complex stress with osmotic and toxic components, which usually predominate in different parts of the plant’s growth cycle. Osmotic is prevalent while plants are producing new tissues, because of the diluting effect of incoming salts. When leaf growth is halted in annual plants, and grain growth dominates, a toxic effect of salt is prevalent. Different mechanisms act under these two stages of salinity stress. Which is the stage of the plants when they are sampled? Are they still producing new leaves and, therefore, still taking advantage of the dilution effect? The authors should put their findings in the context of the stage of plant growth, and comment on the prevalence of osmotic and toxic stresses in their SST treatment. Could you please comment on this?

A: We mentioned in the introduction that salinity stress comprises osmotic and ion toxicity components and considered this in the discussion. The plants were sampled at the vegetative stage of development (32 days after germination), as stated in the Material and Methods section. We do not think that we have observed the dilution effects in growing plants under applied conditions: the incubation medium was fully replaced by fresh every week in order to prevent nutrient depletion and dilution of stress agents. Other words, growing tissues were continuously supplied with nutrients and stress agents over whole experimental time. In this context, the stage of plant growth would not have substantial influence on observed effects.

Classically, osmotic stress due to drought and salinity stress have been compared at similar osmotic stress levels in the root media, caused by different osmotic agents. This seems not the case here (Fig. 1 I), as osmotic stress caused by salinity was much higher than caused by PEG. Can the authors comment on why were the stress levels chosen? Please, justify the validity of comparisons between two stresses which induce so different osmotic stress to the plant. Is it possible to predict what would an osmotic stress caused by PEG, of the same intensity as the one caused by salt, have done? The authors should provide a convincing reply to this concern, to justify their choice, because the comparison of the two types of stress is the backbone of the menuscript.

  1. You are absolutely right that osmotic pressure in two experimental solutions is different. 300 mM NaCl solution is slightly over 500 mOsm or over 12 bars while the 15% PEG6000 solution exhibited nearly 2.5 times lower osmotic pressure. To have equal osmotic pressure, around 32% PEG6000 should be applied. However, our preliminary experiments showed deleterious effects on H. marinum plants after treatment with higher PEG6000 concentrations over selected stress period leading to plant death and material failure for following experiments. Moreover, whereas membrane-permeable Na and Cl Ñ–ons can be used as cheap osmotic agents by plants for osmotic adjustments PEG6000 cannot penetrate plant membranes meaning that PEG6000 may cause more harmful effect. As seen from the phenotypic data, even applied concentration of PEG6000 was more detrimental for plants than high salinity. On contrary, if we would use lower NaCl concentration we could lost some important transcriptomic, metabolomics and physiological changes due to halophytic nature of H. marinum.

It is not clear what new findings are contributed by this study. What is reported reminds of rather similar findings in previous studies. The authors should make an effort to highlight their novel findings, particularly in the intersection of metabolome-transcriptome-physiology, which is the strength of the study.

A: We would like to disagree with this statement. This is the first comprehensive study of the adaptation mechanisms of the halophytic barley H. marinum to salinity which has used physiological, imaging, elemental, metabolic and transcriptomic approaches. Besides, the study provides the first comparative analysis of responses to salinity and osmotic stress in this species. We also showed by NMR that H. marinum roots undergo morphological adjustments in respond to salinity. Candidate genes involved in the adaptation mechanism are also determined. Due to limited space of the manuscript we could not discuss all observations; however, the findings open new ways for research to understand the adaptation mechanisms in more details and to apply those for cultivated crops.

Several aspects of methodology and experimental design need clarification:

L543-544. “The incubation solution was replaced by fresh weekly in order to prevent nutrient depletion” This means that the solution was replaced fully every week? Please clarify the sentence

A: The incubation solution was replaced fully every week. We clarified this sentence in the text.

L548-549. “10-15 plants per hydroponic tank, representing a single replicate of treatment, were harvested, frozen and used in further analyses.” The use of the term “replicate” here is misleading. Was a single replicate used? In Figure 1, n=8, which I understand as 8 biological replicates (probably single plants). Apparently, the number of biological replicates was 8, not 1. Please give a full description of the experimental set-up and experimental design, and clarify the replicates issue, and the nature of the experimental unit (one plant or a bulk of several plants?) Please indicate in the M&M section the number of biological replicates used for each assay: NMR, untargeted metabolite profiling (n=7), mineral analysis (n=8-10), C13 (n=5). State clearly the number of biological reps for RNA-seq: “Three repetitions were performed for each data point”, were this technical or biological?

A: We corrected the description of material sampling and biological/technical replications in the Material and Methods section in order to make them more precisely. Numbers of biological replicates is mentioned either in Figure legends or in M&M section for each experiment.

Also, where all plants of H. marinum and H. vulgare grown in single tanks? Or was there any kind of blocking structure?

A: The H. marinum and H. vulgare were cultivated in separated boxes and did not have any physical contact with each other. Please see Figure 1A and B. We clarified this point in M&M section. 

How many harvests were actually done? L547, the text describes a single harvest, at day 32. But then calculation of RGR (L555) requires at least two harvests, at the beginning and end of the experiment.

A: Yes, we have conducted measurements two times, once at beginning of stress treatment and second one at the end of experiment. However, we did not harvested the plants at the beginning of the treatment but only weighted them and let to grow the same plants till the end of experiment. In this respect, the plants were evaluated twice but harvested only once at the end of experiment for RNAseq, metabolic and ionomic analyses. 

The method to measure leaf osmotic pressure is not explained. This is absolutely needed. The equipment used is not specific for plant tissues. Specific protocols for plant tissues (leaf discs, extracted sap, etc) have been developed (Ball and Oosterhuis, 2005, Env Exp Bot), and used in plant research (for instance Caringella et al 2015, PCE). The authors should describe with enough detail the procedures used, the tissues sampled, etc. Also, I recommend expressing osmotic pressure in the more usual MPa units, to facilitate comparison with other studies.

A: We added more detailed description of the corresponding measurements in M&M section. We recalculated osmotic pressure in more generally used MPa units (former Figure 1I-K, now Figure S1). 

Other issues:

L89 (Figure 1). Not clear what panels I-J-K are

A: We separated and transferred the Figure 1I-K to supplementary materials as Figure S1.

I miss measurement of leaf potential in the report. Was it possible to measure leaf potential in the samples? Please, comment on this matter.

A: The primary aim of the study was evaluation of differences in response to salinity and drought on transcriptional and metabolic levels. We measured osmotic pressure of plant tissue sap of H. marimun and H. vulgare. We have not measured the leaf potential and do not have the plants anymore to perform such measurements.

Is PWC the same as relative water content, which is a trait more commonly found in the literature?

A: PWC is plant water content and differs from the relative water content (RWC) by formula and algorithm of calculation. RWC is more usually used for measurements in leaves. Because we analysed the whole plant, we applied PWC which was calculated as follows: PWC = (FW–DW)/DW, where DW was dry weight, FW was fresh weight of an individual plant. The PWC values were converted into percentage relatively to the value in control condition”. Please see M&M for the description.

In Figure 3, what catches the eye is the much higher uptake of C13 by marimun, compared to vulgare, even at the control treatment. Was this expected? Is there an explanation for this?

A: 13C feeding experiments with H. marinum were conducted for the first time and therefore no reference values existed so far. One possible explanation is higher photosynthetic efficiency  of H. marinum plants. The finding deserves future investigations, which have not been in focus of the present work.

L142. Table 1. Please align numbers in the table. Does colour coding mean anything? In the footnote, include “*t significantly different from the control treatment at…”

A: Colours mean elevation (in blue) or decrease (in red) of element content in the particular plant tissue. We added this statement to the legend.    

L174, please use the term “Principal component analysis”

A: Corrected.

L276. Please, align Table (maybe landscape?)

A: Corrected.

L617. “The P values were adjusted for multiple testing according to Benjamini-Hochberg”. This statement is void unless it is accompanied by more information, like the actual q-value used as cut-off.

A: We corrected this sentence, see lines 619-620.

Evidences of restriction of ion entry and sequestration are not that big (L530).

A: We have provided several indirect evidences that suggest restrictions of ion entry and sequestration. First, the calculated K/Na ratio was much higher in shoots than in roots under SST. Second, K+ transporter SKOR, involved in K+ uploading into the xylem sap, and cation/H+ antiporter, possibly involved in Na+ removal or K+ redistribution, were strongly transcriptionally upregulated in SST-treated roots. Besides, these results are in agreement with previous analyses of H. marinum [18, 58, 66].

Reviewer 3 Report

Good and interesting data, good choice of objects with modern methods and updated methods for processing the results. However, the osmotic components of the stresses do not match each other (Figure 1 I + methods), so the papers should be reconsidered (how to present the good results) + more details for methods to be added. 

1) Figure S1. Circles are distorted in the upper part of the figure, pls, correct.   

2) Abstract. 

When compared with cultivated barley, seaside  

15 barley exhibited superior plant growth rate, relative plant water content, osmotic potential and 16 photosynthetic activity under high salinity, but not under osmotic stress. 

Pls, clarify what were the parameters of osmotic potential, water content etc. Under the salinity (lower, higher etc., “superior” is not a definition here). 

3) Figure 1. A and B, pls, provide the scale with cm, mile, li, inch, ly or so to know the size of the plants. 

4) Figure 1. Pls, indicate how you measured the osmotic pressure in plants (J, K), which parts of the palnts were taken, how the sap was extracted, it’s missing in the methods. 

5) All the figures. The quality of text at the figures 1, 4-7 is relatively low. Pls, consider how to improve it. 

6) Figure 1. The marks at the axis Y are not provided. 

7) Figure 1. The Reviewer admits that osmotic pressure of 300 mM NaCl could be slightly over 500 mOsm or over 12 bars, OK, activity coefficients etc. Why did not you take the same osmotic stress by PEG6000? 

The whole common sense of the paper is vanishing then. Indeed, 15% of PEG6000 are about 5 bars of osmotic pressure (e.g. http://www.plantphysiol.org/content/67/1/64) while salt stress was 12 bars + salt effects. 

How could it then happen that presumably stronger effect of salinity + 2.5 higher osmotic pressure had the same effect on the halophytic barley? 

Evidently, stronger effect was for H. vulgare. 

8) Figure 2. Could you confirm all the data for A-D by anatomical analysis of the corresponding tissues? 

  1. B) E-F What are the % of water content corresponding to colour scales/heat maps?
  2. C) What are the linear scales for the figures?
  3. D) What was the time used to collect the images? Nothing is mentioned in the methods.

9) Figure 3. Where are the bars for statistics for salinity treatment? 

10) Table 1. How did the Authors obtain the plant material? Were all the plants including roots and shoots averaged? Nothing is mentioned in Methods. 

11)  Among ion transporters, the expression of K+ channel SKOR was increased ~11-fold exclusively 320 in SST-treated roots, indicating enhanced re-translocation of K+ as the main Na+ competitor. 

Would be good to provide references for the function of the ion channels, e.g. 

https://pubmed.ncbi.nlm.nih.gov/9741629/ +  

exactly for barley: https://pubmed.ncbi.nlm.nih.gov/12223889/ and similar publications from the labs. 

12) Three repetitions were performed 610 for each data point. The complete data set is deposited at the European Nucleotide Archive with 611 accession number: PRJEB38377.  

https://www.ebi.ac.uk/ena/browser/text-search?query=PRJEB38377 

Not found. 

13) 591 were extracted as described [82], 

Pls, describe more about the samples. 

14) Figure 7c. The present osmotic stress was about 3 times lower than the osmotic component of the salt stress. 

15) Plenty of good interesting data but the whole text should be modified taking into account the existing mismatches. 

Author Response

Respond to the comments of the reviewer 3

Good and interesting data, good choice of objects with modern methods and updated methods for processing the results. However, the osmotic components of the stresses do not match each other (Figure 1 I + methods), so the papers should be reconsidered (how to present the good results) + more details for methods to be added. 

A: Thank you for your commendation and constructive comments. We addressed your critical points as described below in details.

1) Figure S1. Circles are distorted in the upper part of the figure, pls, correct.

A: Corrected.   

2) Abstract. 

When compared with cultivated barley, seaside barley exhibited superior plant growth rate, relative plant water content, osmotic potential and 16 photosynthetic activity under high salinity, but not under osmotic stress. 

Pls, clarify what were the parameters of osmotic potential, water content etc. Under the salinity (lower, higher etc., “superior” is not a definition here). 

A: Corrected. Please see the abstract.

3) Figure 1. A and B, pls, provide the scale with cm, mile, li, inch, ly or so to know the size of the plants. 

A: Corrected. Scale bars are now added to the Figs. 1A, B.

4) Figure 1. Pls, indicate how you measured the osmotic pressure in plants (J, K), which parts of the palnts were taken, how the sap was extracted, it’s missing in the methods.

A: We provided more detailed description of the method in the Material and Methods section. Besides, these figures are recalculated and moved to supplementary material as recommended by the reviewer 1.  

5) All the figures. The quality of text at the figures 1, 4-7 is relatively low. Pls, consider how to improve it. 

A: The low quality of figures was probably due to their conversion into a PFD file. We provided all figures of higher resolution in resubmission.

 6) Figure 1. The marks at the axis Y are not provided.

A: Corrected. 

7) Figure 1. The Reviewer admits that osmotic pressure of 300 mM NaCl could be slightly over 500 mOsm or over 12 bars, OK, activity coefficients etc. Why did not you take the same osmotic stress by PEG6000? 

The whole common sense of the paper is vanishing then. Indeed, 15% of PEG6000 are about 5 bars of osmotic pressure (e.g. http://www.plantphysiol.org/content/67/1/64) while salt stress was 12 bars + salt effects. 

How could it then happen that presumably stronger effect of salinity + 2.5 higher osmotic pressure had the same effect on the halophytic barley? 

Evidently, stronger effect was for H. vulgare. 

  1. You are absolutely right that osmotic pressure in two experimental solutions is different. As you have already mentioned, 300 mM NaCl is slightly over 500 mOsm or over 12 bars while the 15% PEG6000 solution exhibited nearly 2.5 times lower osmotic pressure. To have equal osmotic pressure, around 32% PEG6000 should be applied. However, our preliminary experiments showed deleterious effects on H. marinum plants after treatment with higher PEG6000 concentrations over selected stress period leading to plant death and material failure for following experiments. Moreover, whereas membrane-permeable Na and Cl ions can be used as cheap osmotic agents by plants for osmotic adjustments, PEG6000 cannot penetrate plant membranes meaning that PEG6000 may cause more harmful effect. As seen from the phenotypic data, even applied concentration of PEG6000 was more detrimental for plants than high salinity. On contrary, if we would use lower NaCl concentration we could lost some important transcriptomic, metabolomics and physiological changes due to halophytic nature of H. marinum.

8) Figure 2. Could you confirm all the data for A-D by anatomical analysis of the corresponding tissues? 

  1. B) E-F What are the % of water content corresponding to colour scales/heat maps?
  2. C) What are the linear scales for the figures?
  3. D) What was the time used to collect the images? Nothing is mentioned in the methods.

A: The water signal was measured by MRI, as described M@M and colour code was used to visualize the differences in water content within the living tissues. Translation of relative units into absolute values (e.g. %) can currently not be given but would also not change the differences already shown in colour. This necessary procedure is quite labour-extensive, and would require a new set of experimental work and MRI measurements. This is impossible to perform in a short time and under pandemic situation. Moreover, we provided water content in real units (Figure 1).

We are sorry for the missing scale bars at Figure 2. We added this information to the images.

 9) Figure 3. Where are the bars for statistics for salinity treatment? 

A: Corrected.

 10) Table 1. How did the Authors obtain the plant material? Were all the plants including roots and shoots averaged? Nothing is mentioned in Methods. 

A: All plants were sampled after five days of maximum stress (totally 32 days old) and separated into shoot (crown and growing point) and root (2 cm root tips) fractions. 10-15 plants per hydroponic tank, representing a single biological replicate of treatment, were harvested, frozen and used in further analyses. Thus, averaging comprises mixed plant tissue material from 10-15 plants for every single measurement. Please see the chapter 4.1 for more details. 

11)  Among ion transporters, the expression of K+ channel SKOR was increased ~11-fold exclusively in SST-treated roots, indicating enhanced re-translocation of K+ as the main Na+ competitor. 

 Would be good to provide references for the function of the ion channels, e.g. 

https://pubmed.ncbi.nlm.nih.gov/9741629/ +  

exactly for barley: https://pubmed.ncbi.nlm.nih.gov/12223889/ and similar publications from the labs. 

A: We added suggested references regarding the functions of ion channels.

12) Three repetitions were performed 610 for each data point. The complete data set is deposited at the European Nucleotide Archive with 611 accession number: PRJEB38377. https://www.ebi.ac.uk/ena/browser/text-search?query=PRJEB38377 

Not found.

A: ENA routinely applies the embargo on all the data until they are not yet published. On request, we can send an original e-mail confirming successful data submission. The data will be open after manuscript acceptance.

13) were extracted as described [82], 

Pls, describe more about the samples. 

A: We added detailed description of the sample isolation to M&M section.

 14) Figure 7c. The present osmotic stress was about 3 times lower than the osmotic component of the salt stress. 

A: We included additional description of stress conditions in brackets (300 mM NaCl and 15% PEG6000) to exactly highlight the differences related to the treatments.

 15) Plenty of good interesting data but the whole text should be modified taking into account the existing mismatches. 

A: We appreciate all your comments and have addressed all critical points a

Round 2

Reviewer 2 Report

Comments included in file attached

Author Response

Respond to the comments of the Reviewer 2

Dear Reviewer,

We greatly appreciate your critical evaluation of our manuscript and important suggestions. Our  replay to your comments  is  given below.

There are valid points in these replies. The manuscript would definitely gain if the authors included them in the manuscript as well. The corrections in the manuscript have been minimal.

A: We have made additional efforts to include majority of these points into new version of the manuscript. Our comments below are labeled in bolt italic.

The manuscript presents a thorough analysis of responses of H. Marinum to salt stress. The richness of analytical methods used allows to gain possibly new insights on tolerance mechanisms. I have some concerns about the comparability of these results to the response of H. marinum to of PEG-caused osmotic stress, and the comparison with H. vulgare, which makes a too separate compartment in the manuscript. H. vulgare is included in the research, for comparative purposes, and because barley is an important crop, but this comparison is not carried very far. Most of the work is performed on H. marinum, as acknowledged in the title of the manuscript. The manuscript would have given more useful insights if the comparison with H. vulgare (the cultivated relative), which was limited to morphological and a few physiological measurements, had been extended to NMR, metabolomics and gene expression analyses. Further analyses on the vulgare samples would have provided better insights about the similarities and differences of responses, with implications for plant breeding. Are these analyses still possible? I would appreciate comments on how the mechanisms found in marinum differ from those found in vulgare (in this or in other studies), and if and how mechanisms found in marinum could be imported into vulgare.

In our work, we focused our research on H. marinum, a wild halophytic relative of the cultivated H. vulgare. There is plenty of metabolic and transcriptomic research regarding responses of different H. vulgare cultivars to salinity and osmotic stress (e.g., recently published Osthoff et al. 2019, Yousefirad et al. 2020, Hill et al. 2016, and many others). Although H. vulgare is regarded as relatively salt tolerant when compared with other cultivated Triticeae, cultivated barley experiences still a 55–58% decline in biomass at less than one-third the NaCl concentration in sea water (Colmer et al., 2006). The wild sea-side barley (H. marinum) is capable to grow and reproduce in condition of salt marshes and coastal areas with eventual sea water logging (Garthwaite et al., 2005). H. vulgare belongs to includer type of plants and can accumulate substantial amount of Na in tissues (Mian et al., 2011). The adaptation mechanism of H. marinum to high salinity is expected to be different due to its halophytic nature. The evaluation of these mechanism was a main goal of a study. We intend to transfer the knowledge received in this investigation to H. vulgare by overexpression of some key transporter genes to reach enhanced salinity tolerance.

It is evident that H.vulgare is included in the study to provide the research with a potential application in the future. This is fine. However, I insist that the comparison with H- vulgare is not well exploited in the study. It is unfortunate that simple analyses like element concentrations were not carried out for H. vulgare, This analyses give a good idea of the general reaction of the plant against salt stress. They would have provided a solid ground to compare mechanisms between the two species. This does not mean that what is offered in the manuscript is not worthy of publication, but I think that the authors should take this into account for future research. Still, I believe that the authors could improve the comments on the comparison between the two species, and speculate about which mechanisms/genes that they have found in marinum could be transferable to vulgare, beyond the general statements offered, truly based on new findings of this study. In the revised version, the discussion is untouched.

A: Thanks for your suggestions. We have included comparison between the two species, and suggested which mechanisms/genes, found in H. marinum, could be transferable to H. vulgare. Please see Discussion. However, we want to mention that comparison between these species is only possible on available data (phenotype, 13C uptake, previous publications) to avoid large speculations.

Concluding remarks should be refined to reflect precisely the findings of this study:

L558-559: “To conclude, seaside barley, but not cultivated barley, can tolerate very high salinity levels, (Figure 7C)”. The sentence is misleading and could be easily changed. Fig. 7C is a nice figure on a virtual representation of the salinity tolerance mechanisms at play in H. marinum. It is not a proof of its high salinity tolerance (which are presented in other tables and figures) and, definitely, it shows nothing on H.vulgare.

A: We agree that the sentence was misleading. We have made changes in this sentence. Please see Discussion P. 16.

L559-560: “Seaside barley is likely capable of controlling Na+ and Cl- concentrations in its leaves when its roots are imposed to high salinity” I am not sure that “imposed to” is the right verb. But, more important, unfortunately, chloride concentrations were not measured in this study. It would have been a good idea. It is a cheap determination, routinely done in salinity studies, and it is missed here. The sentence is based on indirect evidences, not a direct results from this study, and should be clarified to avoid misleading readers.

A: We have corrected this sentence and written that our assumption is based on indirect evidences. Please see discussion. P. 14-15. We have changed the verb “imposed to” to “subjected to”.

Authors should acknowledge that salinity stress is actually a complex stress with osmotic and toxic components, which usually predominate in different parts of the plant’s growth cycle. Osmotic is prevalent while plants are producing new tissues, because of the diluting effect of incoming salts. When leaf growth is halted in annual plants, and grain growth dominates, a toxic effect of salt is prevalent. Different mechanisms act under these two stages of salinity stress. Which is the stage of the plants when they are sampled? Are they still producing new leaves and, therefore, still taking advantage of the dilution effect? The authors should put their findings in the context of the stage of plant growth, and comment on the prevalence of osmotic and toxic stresses in their SST treatment. Could you please comment on this?

We mentioned in the introduction that salinity stress comprises osmotic and ion toxicity components and considered this in the discussion. The plants were sampled at the vegetative stage of development (32 days after germination), as stated in the Material and Methods section. We do not think that we have observed the dilution effects in growing plants under applied conditions: the incubation medium was fully replaced by fresh every week in order to prevent nutrient depletion and dilution of stress agents. Other words, growing tissues were continuously supplied with nutrients and stress agents over whole experimental time. In this context, the stage of plant growth would not have substantial influence on observed effects.

I do not agree with the last statement, but I see no point in carrying this argument further.

It is true that the authors acknowledge at the beginning of the introduction that osmotic stress is a part of salt stress, but it is not carried on further. One interesting thing they find is the striking differences between responses to OST and SST in H.marinum, and also in the comparison between species. Vulgare and marinum seem to suffer more or less the same with the OST applied, although the loss of colour in marinum is dramatic, pointing to a higher damage of the photosynthetic apparatus. This is acknowledged in the manuscript. Unfortunately, there were no SPAD measurements of neither species, which would have been very useful in this case. On the other hand, vulgare suffers much more than marinum under SST. I think this is a good example of how different osmotic stress can be in these two different situations. Is vulgare superior to marinum in OST? Could we think of combining mechanisms of both species to achieve a barley highly resilient against drought and salinity?

A: We do not think, that H. vulgare is more tolerant to osmotic stress than H. marinum. Unfortunately, we did not perform transcriptomic study with H. vulgare in our work, but available publications have not supported this idea. H. marinun is typical Mediterranean species capable to survive on harsh environments of marginal and semiarid lands. In contrast, H. vulgare as a cultivated crop has more demands in terms of water usage and more sensitive to drought. In this context, it would a good option to study further the mechanisms of H. marinum osmotic tolerance with different range of osmotic agents and concentrations and compare them with H. vulgare.

Besides, these two species have relatively different strategies to overcome salinity stress. H. vulgare is considered as ‘includer’ type and can accumulate large Na+ quantities in green tissues (Mian et 2011). Therefore, it would not be favorable strategy for plants to combat high salinity. On contrary, H. marinum demonstrates capability to control Na+ flux to the shoots. Therefore, it would be very intriguing to transfer some genes from H. marinum to H. vulgare (key of Na+/K+ or Cl+ transporters as well as some sugar transporters and ROX detoxification genes). We have included several suggestions of such candidate genes in Discussion.    

Classically, osmotic stress due to drought and salinity stress have been compared at similar osmotic stress levels in the root media, caused by different osmotic agents. This seems not the case here (Fig. 1 I), as osmotic stress caused by salinity was much higher than caused by PEG. Can the authors comment on why were the stress levels chosen? Please, justify the validity of comparisons between two stresses which induce so different osmotic stress to the plant. Is it possible to predict what would an osmotic stress caused by PEG, of the same intensity as the one caused by salt, have done? The authors should provide a convincing reply to this concern, to justify their choice, because the comparison of the two types of stress is the backbone of the menuscript.

You are absolutely right that osmotic pressure in two experimental solutions is different. 300 mM NaCl solution is slightly over 500 mOsm or over 12 bars while the 15% PEG6000 solution exhibited nearly 2.5 times lower osmotic pressure. To have equal osmotic pressure, around 32% PEG6000 should be applied. However, our preliminary experiments showed deleterious effects on H. marinum plants after treatment with higher PEG6000 concentrations over selected stress period leading to plant death and material failure for following experiments. Moreover, whereas membrane-permeable Na and Cl Ñ–ons can be used as cheap osmotic agents by plants for osmotic adjustments PEG6000 cannot penetrate plant membranes meaning that PEG6000 may cause more harmful effect. As seen from the phenotypic data, even applied concentration of PEG6000 was more detrimental for plants than high salinity. On contrary, if we would use lower NaCl concentration we could lost some important transcriptomic, metabolomics and physiological changes due to halophytic nature of H. marinum.

These explanations are fine, stress levels were chosen based on practical issues and results of preliminary experiments. OST applied was as high as H. marinum could tolerate, and SST level was sensibly high, to elicit salt adaptation responses in H- marinum. This reasoning is not bullet proof, but is valid and should be included in the manuscript.

A: We appreciate your comment. We have added additional description to the manuscript (Pp. 2, 14).

It is not clear what new findings are contributed by this study. What is reported reminds of rather similar findings in previous studies. The authors should make an effort to highlight their novel findings, particularly in the intersection of metabolome-transcriptome-physiology, which is the strength of the study.

A: We would like to disagree with this statement. This is the first comprehensive study of the adaptation mechanisms of the halophytic barley H. marinum to salinity which has used physiological, imaging, elemental, metabolic and transcriptomic approaches. Besides, the study provides the first comparative analysis of responses to salinity and osmotic stress in this species. We also showed by NMR that H. marinum roots undergo morphological adjustments in respond to salinity. Candidate genes involved in the adaptation mechanism are also determined. Due to limited space of the manuscript we could not discuss all observations; however, the findings open new ways for research to understand the adaptation mechanisms in more details and to apply those for cultivated crops.

I think there was a misunderstanding here. I was not implying that there were no new findings. Rather, that the new findings were not highlighted enough, and still are not.

A. We have included several sentences highlighting new findings of our study by rewriting some points in Discussion.

Several aspects of methodology and experimental design need clarification:

L543-544. “The incubation solution was replaced by fresh weekly in order to prevent nutrient depletion” This means that the solution was replaced fully every week? Please clarify the sentence

A: The incubation solution was replaced fully every week. We clarified this sentence in the text.

L548-549. “10-15 plants per hydroponic tank, representing a single replicate of treatment, were harvested, frozen and used in further analyses.” The use of the term “replicate” here is misleading. Was a single replicate used? In Figure 1, n=8, which I understand as 8 biological replicates (probably single plants). Apparently, the number of biological replicates was 8, not 1. Please give a full description of the experimental set-up and experimental design, and clarify the replicates issue, and the nature of the experimental unit (one plant or a bulk of several plants?) Please indicate in the M&M section the number of biological replicates used for each assay: NMR, untargeted metabolite profiling (n=7), mineral analysis (n=8-10), C13 (n=5). State clearly the number of biological reps for RNA-seq: “Three repetitions were performed for each data point”, were this technical or biological?

A: We corrected the description of material sampling and biological/technical replications in the Material and Methods section in order to make them more precisely. Numbers of biological replicates is mentioned either in Figure legends or in M&M section for each experiment.

The minimum correction added in line 18, page 16, actually does not help. The biological replicate in this experiment is a single plant. Three, 5, 8 plants (or biological replicates) were used for different purposes. Please correct the sentence: “10-15 plants per hydroponic tank, representing a single biological replicate of treatment, were harvested, frozen and used in further analyses.” Something along these lines: Each treatment was applied in separate hydroponic tanks. In each tank, 10-15 plants of H. vulgare or H. marinum were grown. For all determinations, each plant was considered a biological replicate.

A: We believe, there is a misunderstanding in this point. A single biological replication in our experiment was not a single plant but a bulk of 10-15 plants in order to avoid possible genetic and environmental influences, which may be associated with single plants. We have rewritten this paragraph more clearly.

The following sentence must be corrected. Because “fresh” is an adjective, it needs to go with a noun: “The incubation solution was fully replaced by fresh weekly in order to prevent nutrient depletion”

A: We have exchanged word “fresh” on “newly prepared”.

Also, where all plants of H. marinum and H. vulgare grown in single tanks? Or was there any kind of blocking structure?

A: The H. marinum and H. vulgare were cultivated in separated boxes and did not have any physical contact with each other. Please see Figure 1A and B. We clarified this point in M&M section.

How many harvests were actually done? L547, the text describes a single harvest, at day 32. But then calculation of RGR (L555) requires at least two harvests, at the beginning and end of the experiment.

A: Yes, we have conducted measurements two times, once at beginning of stress treatment and second one at the end of experiment. However, we did not harvested the plants at the beginning of the treatment but only weighted them and let to grow the same plants till the end of experiment. In this respect, the plants were evaluated twice but harvested only once at the end of experiment for RNAseq, metabolic and ionomic analyses.

Ok, please reflect this important point in the manuscript.

The method to measure leaf osmotic pressure is not explained. This is absolutely needed. The equipment used is not specific for plant tissues. Specific protocols for plant tissues (leaf discs, extracted sap, etc) have been developed (Ball and Oosterhuis, 2005, Env Exp Bot), and used in plant research (for instance Caringella et al 2015, PCE). The authors should describe with enough detail the procedures used, the tissues sampled, etc. Also, I recommend expressing osmotic pressure in the more usual MPa units, to facilitate comparison with other studies.

A: We added more detailed description of the corresponding measurements in M&M section. We recalculated osmotic pressure in more generally used MPa units (former Figure 1I-K, now Figure S1).

Ok

Other issues:

L89 (Figure 1). Not clear what panels I-J-K are

A: We separated and transferred the Figure 1I-K to supplementary materials as Figure S1.

Moving panels I-J-K to a supplementary figure is hiding, not clarifying. Maybe an improved legend would have sufficed.

A: We have improved legend for this figure and included more information regarding measurements. Please see Figure S1. However, we prefer to leave it in the supplementary material due to the overloading of Figure 1.

I miss measurement of leaf potential in the report. Was it possible to measure leaf potential in the samples? Please, comment on this matter.

A: The primary aim of the study was evaluation of differences in response to salinity and drought on transcriptional and metabolic levels. We measured osmotic pressure of plant tissue sap of H. marimun and H. vulgare. We have not measured the leaf potential and do not have the plants anymore to perform such measurements.

Ok, this is unfortunate. Maybe you should consider that for future experiments.

A. We have realized the importance to measure leave potential and photosynthetic activity for future experiments.

Is PWC the same as relative water content, which is a trait more commonly found in the literature?

A: PWC is plant water content and differs from the relative water content (RWC) by formula and algorithm of calculation. RWC is more usually used for measurements in leaves. Because we analysed the whole plant, we applied PWC which was calculated as follows: PWC = (FW–DW)/DW, where DW was dry weight, FW was fresh weight of an individual plant. The PWC values were converted into percentage relatively to the value in control condition”. Please see M&M for the description.

Ok

In Figure 3, what catches the eye is the much higher uptake of C13 by marimun, compared to vulgare, even at the control treatment. Was this expected? Is there an explanation for this?

A: 13C feeding experiments with H. marinum were conducted for the first time and therefore no reference values existed so far. One possible explanation is higher photosynthetic efficiency of H. marinum plants. The finding deserves future investigations, which have not been in focus of the present work.

Ok, please include this comment in the manuscript, as it may interest the readers

A: We have included this information in Results (P. 4) and Discussion (P. 14).

L617. “The P values were adjusted for multiple testing according to Benjamini-Hochberg”. This statement is void unless it is accompanied by more information, like the actual q-value used as cut-off.

A: We corrected this sentence, see lines 619-620.

I have not found that correction in the manuscript. Also, lines are numbered differently.

A: We have corrected it. Unfortunately, line numbering was changed during formatting of last version.

The K/Na ratio was higher in shoots than in roots, both in SST and in the control treatment (though the ratio in the last case was much higher due to the low presence of Na in the medium). Additionally, the authors state that “Even though Na+ was highly accumulated in SST-treated tissues, its concentration in the shoots was 1.4 fold lower than in the roots, implicating active Na+ recruitment as an additional cheap osmotic agent in the latter, and suggesting prevention of Na+ transport to photosynthetic tissues.”. But the ratio root-to shoot for Na in the control treatment was very close to that value, 1.35. These facts indicate that, if Na is excluded from the shoots (again, it would be nice to have H. vulgare values to compare), and K is favoured, these phenomena may occur constitutively and, therefore, may not result in gene expression changes. I am not saying this was the case, but I would like.

A. We have included in discussion data form previous research regarding H. vulgare Na+ content in different plant tissues. Our data about K+/N+ ratio strongly correlate with data obtained by other researchers [18, 20, 59, 70] and differ from that of H. vulgare. Please see Discussion P. 15. If consider high amount of Na+ in the incubation medium, it is very unlikely that H. marinum plants could capable to cope with high salinity employing only constitutive mechanisms. Our data revealed significant changes in membrane transport transcription profile as well as substantial metabolic and morphologic adjustments.

In this respect, I would appreciate some more elaboration on the role of K. P5,L11-13. “Contrary to Na, the K and particularly the Ca contents were significantly reduced in the shoots and roots of SST-treated plants. “ The reduction of K under SST was just an effect of passive uptake due to the reduced concentration of K in the root medium, compared to Na? Is there an indication of active uptake and translocation of K, in a medium far richer in Na?

A: The elevation of Na+ and reduction of Ca2+ could be explained by Ca2+ replacement by Na+ in cell walls and vacuoles, the compartments with the largest Ca2+ pools in plants. K+ is considered as a main competitor of Na+ in uptake and transport. Because K+ is substantial element required for photosynthesis and cell metabolism, enrichment of plant tissues by K+ would minimize the toxic effects of Na+. To our opinion, activation of HmSKOR plays crucial role in long distance K+ transport, the idea we are going to analyze in future. K+ reduction observed in SST plants and OST roots could be caused by stress-induced K+ leakage, observed for my stress types. We have made corresponding changes in Discussion to explain this topic more precisely.

Reviewer 3 Report

The Reviewer would like to stress again that the obtained results are interesting and supported by a large number of modern methods.

However, still there are two major points which are rather hidden now in the submitted MS for the potential readers of the text.

Osmotic stress applied (by PEG) was 3 times lower in bars that the salt stress involved. So, presumably the comparisons are not valid.

However, the growth response of plants was similar, PEG is not permeable, NaCl is permeable. Probably it’s the only way to equalise the osmotic and salt stress given.

The osmotic pressures of sap from plants were quite different after the stresses, so still the problem remains but it’s rather a general problem, not for the Authors.

Ideally, it should be the other 2-3 sets of experiments with 2-3 extra conditions (similar osmotic stress from PEG and NaCl, low and higher, range of concentrations of NaCl) but it is then too far for the paper for now with metabolomics and transcriptomics + MRI.

MRI imaging, what are the red – blue differences for the scale at Figure 2, E-F, 1%, 50%, 10%, at least any indication is needed (no matter if it’s a pandemic situation or not).

The Reviewer suggests

1) to add the parts of the replies to the text that osmotic stress by PEG (it’s a very specific impermeable compound, so not so strange that the osmotic effects are too strong) was similar in growth responses for H. marinum

and

2) to indicate somehow the quantitative meaning of the heat map scale for MRI imaging (Figure 2 E-F). Will the paper be amended or removed after the pandemic situation?

+ to add what the units are for the linear scale given now at Figure 2.

7) Figure 1. The Reviewer admits that osmotic pressure of 300 mM NaCl could be slightly over 500 mOsm or over 12 bars, OK, activity coefficients etc. Why did not you take the same osmotic stress by PEG6000? 

The whole common sense of the paper is vanishing then. Indeed, 15% of PEG6000 are about 5 bars of osmotic pressure (e.g. http://www.plantphysiol.org/content/67/1/64) while salt stress was 12 bars + salt effects. 

How could it then happen that presumably stronger effect of salinity + 2.5 higher osmotic pressure had the same effect on the halophytic barley? 

Evidently, stronger effect was for H. vulgare. 

  1. You are absolutely right that osmotic pressure in two experimental solutions is different. As you have already mentioned, 300 mM NaCl is slightly over 500 mOsm or over 12 bars while the 15% PEG6000 solution exhibited nearly 2.5 times lower osmotic pressure. To have equal osmotic pressure, around 32% PEG6000 should be applied. However, our preliminary experiments showed deleterious effects on H. marinum plants after treatment with higher PEG6000 concentrations over selected stress period leading to plant death and material failure for following experiments. Moreover, whereas membrane-permeable Na and Cl ions can be used as cheap osmotic agents by plants for osmotic adjustments, PEG6000 cannot penetrate plant membranes meaning that PEG6000 may cause more harmful effect. As seen from the phenotypic data, even applied concentration of PEG6000 was more detrimental for plants than high salinity. On contrary, if we would use lower NaCl concentration we could lost some important transcriptomic, metabolomics and physiological changes due to halophytic nature of H. marinum.

8) Figure 2. Could you confirm all the data for A-D by anatomical analysis of the corresponding tissues? 

  1. B) E-F What are the % of water content corresponding to colour scales/heat maps?
  2. C) What are the linear scales for the figures?
  3. D) What was the time used to collect the images? Nothing is mentioned in the methods.

A: The water signal was measured by MRI, as described M@M and colour code was used to visualize the differences in water content within the living tissues. Translation of relative units into absolute values (e.g. %) can currently not be given but would also not change the differences already shown in colour. This necessary procedure is quite labour-extensive, and would require a new set of experimental work and MRI measurements. This is impossible to perform in a short time and under pandemic situation. Moreover, we provided water content in real units (Figure 1).

We are sorry for the missing scale bars at Figure 2. We added this information to the images.

 14) Figure 7c. The present osmotic stress was about 3 times lower than the osmotic component of the salt stress. 

A: We included additional description of stress conditions in brackets (300 mM NaCl and 15% PEG6000) to exactly highlight the differences related to the treatments.

Author Response

Respond to the second review of the Reviewer 3

Comments and Suggestions for Authors

The Reviewer would like to stress again that the obtained results are interesting and supported by a large number of modern methods.

However, still there are two major points which are rather hidden now in the submitted MS for the potential readers of the text.

Osmotic stress applied (by PEG) was 3 times lower in bars that the salt stress involved. So, presumably the comparisons are not valid.

However, the growth response of plants was similar, PEG is not permeable, NaCl is permeable. Probably it’s the only way to equalise the osmotic and salt stress given.

The osmotic pressures of sap from plants were quite different after the stresses, so still the problem remains but it’s rather a general problem, not for the Authors.

Ideally, it should be the other 2-3 sets of experiments with 2-3 extra conditions (similar osmotic stress from PEG and NaCl, low and higher, range of concentrations of NaCl) but it is then too far for the paper for now with metabolomics and transcriptomics + MRI.

MRI imaging, what are the red – blue differences for the scale at Figure 2, E-F, 1%, 50%, 10%, at least any indication is needed (no matter if it’s a pandemic situation or not).

The Reviewer suggests

1) to add the parts of the replies to the text that osmotic stress by PEG (it’s a very specific impermeable compound, so not so strange that the osmotic effects are too strong) was similar in growth responses for H. marinum

A: We have added the information about different osmotic pressures, achieved by either of stress, to Results (P. 2) and mentioned it in Discussion (P. 14). 

2) to indicate somehow the quantitative meaning of the heat map scale for MRI imaging (Figure 2 E-F). Will the paper be amended or removed after the pandemic situation?

+ to add what the units are for the linear scale given now at Figure 2.

A: The MRI experiment was planned and conducted in such way, which allowed visualization of the relative differences of water distribution in leaving roots. Internal calibration of MRI equipment is required to receive the quantification of data. The quantitative differences in root water content (in %) is shown at the Figure 1. MRI signal was represented by rainbow-based color scale, identical for control and stressed roots. We have performed changes to the legend of Figure 2 with more detailed description. 

Round 3

Reviewer 2 Report

The authors have modified the manuscript following the suggestions made in the last review round. Some minor language edits may be needed but in my opinion, the manuscript is acceptable for publication.

Author Response

The authors have modified the manuscript following the suggestions made in the last review round. Some minor language edits may be needed but in my opinion, the manuscript is acceptable for publication.

A. We have shown the latest version of the manuscript to the English-spoken person and performed minor language corrections upon recommendations. Please see labelled positions in the submitted manuscript.

Reviewer 3 Report

The Reviewer is basically satisfied by the changes concerning the osmotic pressures of external solution by PEG6000 and by the applied solution for salt stress (300 mM NaCl).

Potentially it would be good to discuss the reasons for the situation (indeed, impermeable PEG6000 is quite different from NaCl which could be used by halophyte for osmoregulation).

The Reviewer would prefer to see at least a few phrases in the discussion concerning the osmoregulation mechanisms revealed.

Indeed, the measured concentrations of Na+ after the salt treatment were not that huge as it would be expected from the osmotic pressure.

The concentrations (Table 1 with ions) would correspond to about 150 mM, that is below 1 MPa in osmotic pressure. What would make the remaining 7 MPa (based on figure 1s)? The Reviewer assumes that the observation and further research in the direction could be of interest.

Table 1s does not indicate to a specific compound which could be responsible for such the high changes in osmotic pressure.

Unfortunately, the Reviewer is less positive about the MRI experiments in the described context. Using MRI instead of simple methods of morphology is not the great advantage of the submitted text.

Surprisingly and impressively strangely, the lowest water in the technically advanced experiments was shown for water carrying xylem and phloem (Figure 2E, F). Definitely, the scale of water content apart from the heat map scale is needed or MRI results otherwise the data do not present any new knowledge.

Basically, the volume of useful information is sufficient for a publication though the above mentioned points are remaining for the Authors.

Respond to the second review of the Reviewer 3

Comments and Suggestions for Authors

The Reviewer would like to stress again that the obtained results are interesting and supported by a large number of modern methods.

However, still there are two major points which are rather hidden now in the submitted MS for the potential readers of the text.

Osmotic stress applied (by PEG) was 3 times lower in bars that the salt stress involved. So, presumably the comparisons are not valid.

However, the growth response of plants was similar, PEG is not permeable, NaCl is permeable. Probably it’s the only way to equalise the osmotic and salt stress given.

The osmotic pressures of sap from plants were quite different after the stresses, so still the problem remains but it’s rather a general problem, not for the Authors.

Ideally, it should be the other 2-3 sets of experiments with 2-3 extra conditions (similar osmotic stress from PEG and NaCl, low and higher, range of concentrations of NaCl) but it is then too far for the paper for now with metabolomics and transcriptomics + MRI.

MRI imaging, what are the red – blue differences for the scale at Figure 2, E-F, 1%, 50%, 10%, at least any indication is needed (no matter if it’s a pandemic situation or not).

The Reviewer suggests

1) to add the parts of the replies to the text that osmotic stress by PEG (it’s a very specific impermeable compound, so not so strange that the osmotic effects are too strong) was similar in growth responses for H. marinum

A: We have added the information about different osmotic pressures, achieved by either of stress, to Results (P. 2) and mentioned it in Discussion (P. 14).

2) to indicate somehow the quantitative meaning of the heat map scale for MRI imaging (Figure 2 E-F). Will the paper be amended or removed after the pandemic situation?

+ to add what the units are for the linear scale given now at Figure 2.

A: The MRI experiment was planned and conducted in such way, which allowed visualization

of the relative differences of water distribution in leaving roots. Internal calibration of MRI equipment is required to receive the quantification of data. The quantitative differences in root water content (in %) is shown at the Figure 1. MRI signal was represented by rainbow-based color scale, identical for control and stressed roots. We have performed changes to the legend of Figure 2 with more detailed description.

Author Response

Dear Reviewer,

We greatly appreciate your time and you suggstions and remarks.  Our  replay to your comments  is  given below.

The Reviewer is basically satisfied by the changes concerning the osmotic pressures of external solution by PEG6000 and by the applied solution for salt stress (300 mM NaCl).

Potentially it would be good to discuss the reasons for the situation (indeed, impermeable PEG6000 is quite different from NaCl which could be used by halophyte for osmoregulation).

 The Reviewer would prefer to see at least a few phrases in the discussion concerning the osmoregulation mechanisms revealed.

A: We have inserted additional information to the discussion regarding possible mechanisms of osmoregulation as suggested by the reviewer. Please see Discussion, p. 14, lines 29-33.

Indeed, the measured concentrations of Na+ after the salt treatment were not that huge as it would be expected from the osmotic pressure.

The concentrations (Table 1 with ions) would correspond to about 150 mM, that is below 1 MPa in osmotic pressure. What would make the remaining 7 MPa (based on figure 1s)? The Reviewer assumes that the observation and further research in the direction could be of interest.

Table 1s does not indicate to a specific compound which could be responsible for such the high changes in osmotic pressure.

A: The osmotic adjustment by ions is just one part of the complex mechanism of osmoregulation in H. marinum. Also, we have measured just a fixed range of elements, mainly belonging to the cation component. Unfortunately, we did not have the opportunity to measure the main anions. We believe that anions, in particular Cl-, would substantially contribute to the plant osmotic pressure. It worth to mention that osmotic pressure is built up in plants not only by ions but by other types of osmolytes including sugars, amino acids and ureides. This part of the osmotic adjustment is discussed in detail in the discussion. Please see Discussion, p. 14, lines 34-45. To sum up, the ion concentrations from Table 1 and the osmotic pressure (below 1 MPa), calculated from them, represents just a part of the “whole pie”. Besides the ions, H. marinum plants actively use proline, mannitol, trehalose and ureide for osmotic adjustments. Thus, we cannot highlight the specific compound responsible for such high changes in osmotic pressure because the mechanism of osmotic adjustment in these plants is very complex and comprises a broad range of components.

Unfortunately, the Reviewer is less positive about the MRI experiments in the described context. Using MRI instead of simple methods of morphology is not the great advantage of the submitted text.

Surprisingly and impressively strangely, the lowest water in the technically advanced experiments was shown for water carrying xylem and phloem (Figure 2E, F). Definitely, the scale of water content apart from the heat map scale is needed or MRI results otherwise the data do not present any new knowledge.

Basically, the volume of useful information is sufficient for a publication though the above mentioned points are remaining for the Authors.

A: We appreciate your valuable attention to MRI. As you admitted, the MRI correctly reflects the differences (quantified in Fig. 1) in vivo, and yes (!) it raises more questions. This explains why we bring MRI "in the described context" to this special issue: MRI provides alternative validation and, at the same time, opens new perspectives for the investigator. As mentioned earlier, we can currently not give more details (absolute scales of water content). Nevertheless, the possibility of noninvasive observations which we demonstrate here is generally of advantage. The method allows for non-destructive analysis (and even to observe the dynamic of changes during growth or interactions). Of course, one could go into more detail using a microscopic level of resolution or absolute scales.  We still hope it will attract the attention of specialists in the field, and that it will initiate the development of more precise and advanced application.